# Securing digital images: A chaos-driven scrambling algorithm using the Rössler system

Nasreen Zulfiqar[1], Tauqir Ahmad[1], Taher M. Ghazal[2†], Atif Ikram[3,4†*], Foziah Gazzawe[5†], Marwan Ali Albahar[6‡], Abdelrahman H. Hussein[2‡], Ahmad Salman Khan[7‡]

**1** Department of Computer Science, UET, Lahore, Pakistan, **2** Department of Networks and Cybersecurity, Hourani Center for Applied Scientific Research, Al-Ahliyya Amman University, Amman, Jordan, **3** Department of Computer Science & IT, The University of Lahore, Lahore, Pakistan, **4** Faculty of Computer Science and Multimedia, Lincoln University College, Selangor Darul Ehsan, Malaysia, **5** Computer Science department, Umm Al-Qura University, Mecca, Saudi Arabia, **6** Department of Computing, College of Engineering and Computing in Al-Lith, Umm Al-Qura University, Mecca, Saudi Arabia, **7** Department of Software Engineering, The University of Lahore, Lahore, Pakistan

☯ These authors contributed equally to this work.
† These authors contributed equally to this work.
‡ These authors contributed equally to this work.

* aikram4u@gmail.com

**Data availability statement:** All relevant data are within the manuscript.

## Abstract

A wide range of mathematical constructs, theories, and algorithms have been leveraged by cryptographers to secure valuable and sensitive digital images. Among these, image encryption based on scrambling algorithms has been extensively utilized. These scrambling techniques operate on 1D, 2D, and 3D constructs, relying on pixel-level swapping to enhance security. This research introduces a novel scrambling algorithm that employs a row- and column-based scrambling approach. The encryption process begins with an input image, from which a 2D scrambled image is generated. The columns of pixels from the input image are randomly placed into the columns of the scrambled image. Once all columns are inserted, the process is repeated to intensify the scrambling effect. A similar procedure is then applied to the rows of the image. To further enhance security, a diffusion process is applied using an XOR operation between the scrambled image and a random number stream. These random numbers are generated using the 3D Rössler chaotic system, where two chaotic streams contribute to the scrambling effect, and a third stream is used for diffusion. Both machine simulations and exhaustive security analyses demonstrate that the proposed image cipher is highly resistant to various cryptanalytic attacks, including brute-force attacks, differential attacks, and noise and data cropping attacks. Given these promising results, we recommend the proposed method for real-world applications to fully utilize its security potential.

**Funding:** The author(s) received no specific funding for this work.

# 1 Introduction

In an era characterized by the continuous storage and transmission of digital data, ensuring the integrity, privacy and confidentiality of this data has become very important and urgent. The niche of image encryption—a crucial subset of data security, carries out the protection of visual information from the illegitimate access and tempering. Since digital images are ubiquitous and they are perennially being used in the various fields like medicine, social media, traffic, research, data science to name a few, development of robust image encryption techniques has become paramount. Traditionally, the ciphers like Data Encryption Standard (DES), Advanced Encryption Standard (AES), International Data Encryption Algorithm (IDEA) and Rivest-Shamir-Adleman (RSA) have been employed by the industry experts, practitioners and academicians to safeguard their data during the storage and transmission. One of the limitations of these ciphers is that they can carry out the job of encryption on the textual data. They can't be used for the digital images. Hence, the cryptographers have paid an increasing attentions towards exploring the different techniques for the image encryption.

Varied chaotic maps and other random number generators have rendered a great job in spawning the streams of random numbers. These random numbers are fundamentally raw and primitive in their character and orientation. So, normally, they are customized and normalized in a particular range of the values so that they can be employed over the pixels' data of the given plaintext images. These random numbers carry out the two cardinal operations of confusion and diffusion lying at the very heart of the enterprise of cryptography. In confusion, the pixels are physically disordered and permuted. In the diffusion, their values are changed.

Many ciphers for the images have been produced during the last two decades or so for the security of digital images [8–10,51–53,56,57,65,66,76]. The instruments of chaotic maps [14–17] and DNA encoding [18–21] have been a part of the majority of these algorithms. Besides, other mathematical constructs have also been employed to realize the effects of scrambling like nine palace [22], king [23], cellular automata [24], 15-puzzle [25] latin square [26] and latin cube [27], Kenken puzzle [28], Castle [29], etc. In [22], the Cyclic Redundancy Check (CRC) and nine palace map were employed to come up with a secured image cipher. The theory of the nine palace was exploited to shuffle the color image pixels in given plain image. Besides, bits of three channels were cyclically shifted using the CRC code. A random key matrix was used to get the effects of diffusion. An entropy of 7.9993 was obtained. In another work [29], the chess piece Castle was employed to get the confusion effects. This Castle randomly walked on the large hypothetical chessboard. As it walked, the pixels taken from the given image were migrated to the random positions of the other image. To introduce more security effects, DNA encoding was also used.

An other work [33] has developed a novel scheme for image encryption in which an interdependence between encrypted planes has been carried out. In particular, it reduced encryption time and enhanced the security by embedding a complicated XoR operation. The reported work used a substitution technique that employed XoR operations among pixels. This action was inspired by fast Walsh-Hadamard transform algorithm. Moreover, encryption has many key phases which enhanced efficiency and security of the system. Firstly, line processing involved fusing lines from varied planes and application of chaotic substitution permutation operations. Secondly, the same kind of operation was done to the columns of the given image. Lastly, the planes were broken down into the intersecting squared sub-blocks along with an XoR chaotic confusion operation applied simultaneously to three-channel sub-blocks. Besides, the work [34] not only presents an image encryption algorithm but also a novel chaotic system named as fractional-order five-dimensional hyper-chaotic system

(F5DHS). Apart from that, this work also used an extended DNA encoding scheme to spawn more DNA encoding rules. Additionally, four DNA computation methods were also used to heighten the security of the potential encryption algorithm. Moreover, a blocked image encryption method was engineered to distribute the RGB image into sub-blocks. Later on, DNA encoding, DNA decoding and DNA arithmetic rules were determined for sub-blocks basedon the chaotic data.

Image encryption based on the scrambled images is also one of the most used techniques employed by the image cryptographers. These techniques include 1D scrambled image [35] 2D scrambled image [36,37] and 3D [38] etc. The particular modus oprandi of these techniques works like this. Initially, scrambled image is generated. The pixels values of these images is normally initialized by some value say −1. Then pixels' data from the given image is inserted one by one on the different cells of the scrambled image. During this process, sometimes the data can't be inserted due to the pixels' cells which are already occupied. Due to this reason, the remaining data is inserted on the vacant locations of the scrambled image. In this way, the entire data of the given plain image is inserted in the scrambled image. This is how the scrambling effects are realized in this type of technique. The study [35] created a 1D scrambled image. Then the pixels data from the given plain image were inserted on the different 1D positions of the scrambled image. Random numbers generated by the 2D logistic chaotic map were utilized in this process. Besides, the work [37] created a 2D scrambled image. Moreover, the reported work employed the Langton's ant for transferring the pixels from the given plain image to the different locations of the scrambled image. In the same fashion, the work [38] extrapolated this concept to the three dimensions. First of all, a 3D space was created, now the pixels data from the given color image was inserted to the different positions of the 3D space. In all these works, chaotic maps were employed to generate the streams of random data.

Some image ciphers got broken by the cause of numerous security defects in their design methodologies [5,6]. As an example, the work [30] was cracked by [31] due to the lack of plaintext feature in its design principle. Moreover, some ciphers happen to be very time-consuming in their decryption and encryption algorithms. For instance, the work [22] took a time of around 20 seconds for encryption and decryption of the given digital images which is not in line with the urgent requirements of the current times. Hence, remedial measures must be adopted to thwart future hackers and other adversaries and to design the speedier image ciphers.

Inspired by the above discussion, in particular of [35–38], this study has rendered a yet another image encryption algorithm. First of all, a 2D scrambled image is created. Then the randomly chosen rows from the give image are inserted to the different rows of the 2D scrambled image. This process has been iterated numerously to satisfactorily inject the scrambling effects. The same operation has been carried out based on the columns of the given plain image. 3D Rössler chaotic system [13] has been employed to generate the streams of random numbers. A simple but powerful XoR operation has been carried out to embed the diffusion effects in the proposed image cipher. The following bullets objectively describe the contributions of the current research endeavor.

- A novel image encryption algorithm has been proposed that employs iterative row and column insertion into a 2D scrambled image to achieve strong confusion effects.
- A 3D Rössler chaotic system has been used to generate high-quality random sequences that drive the scrambling process, enhancing unpredictability and security.
- A lightweight yet effective XOR-based diffusion mechanism has been integrated, ensuring a robust transformation of the plaintext image into a secure ciphertext.

The rest of the paper has been formatted like this. Section 3 describes the necessary concept of Rössler chaotic map/system. In Section 4, the proposed encryption scheme for the digital images has been presented in detail. Machine experimentation along with other performance evaluations have been given in Sections 5 and 6. Finally, the Section 7 closes the article with requisite concluding remarks and possible future research directions.

## 2 Related work

Due to the dramatic rise of the digital images in the current era of time, cryptographers paid an increasing attention to their security. Here, we will describe few of the published works.

With the focus on the encryption of medical image, a research [67] utilized chaotic systems in their encryption algorithm. In the start, there was a key generation layer. Next, there were 2 encryption rounds in which DNA computing and chaotic systems were amalgamated to harness their fuller potential. All of this was done by complying with the classical cryptographic structure of permutation, substitution and diffusion. The secret parameters of chaotic systems were generated through the initial parameters alongside the SHA-256 hash function. The algorithm passed through finite number of rounds. Each round consisted of six steps, namely permutation on block-based system, substitution on pixel-based system, DNA encoding, substitution on bit-level (which is DNA complementing), DNA decoding, and diffusion on bit-level system. Using the logistic-Chebyshev map during the substitution of bit-level system, key strings were generated whereas the diffusion of bit-level system helped achieve key strings utilizing sine-Chebyshev map. The repetition of the reported steps along with the secret keys resulted in the required cipher image.

A research study [68] suggested a novel 1D fractional chaotic mapping. Furthermore, the reported work also suggested DNA encoding, parallelly utilizing chaotic mapping. The combination of fraction operation and sine map rendered a yet another chaotic map. The suggested algorithm also exploited SHA-3 hash codes to resist potential plaintext attack. In another work [69], a new 1D SCCM (sine - cosine chaotic map) was proposed to enhance the cryptographic strength of the existing 1D chaotic map. The SCCM when conjoined with DNA operations produced an image encryption algorithm. For the confusion phase, columns and rows of the input image were relocated and this was done through the chaotic sequence achieved through SCCM. Following the rule of "one pixel for one rule," chaotic sequences helped select the coding, decoding and calculating rules as suggested by the random DNA coding. SHA-512 hash helped generate the initial parameters of the chaotic mapping.

The combined use of the chaotic systems and DNA sequence operations resulted in another image encryption algorithm [70]. The cryptographic structure of traditional permutation and diffusion was followed in this work. The initial system parameters were calculated using 256-bit hash function. These parameters were used for 2D LASM (two-dimensional Logistic adjusted Sine map) along with a new one-dimensional chaotic mapping. The 2D LASM produced the chaotic sequence which was utilized in rule matrix of DNA encoding and DNA decoding; this encoded the input plain image in the DNA matrix accordingly. The permutation of columns and rows was done on DNA level at the DNA matrix of the original plain image. Apart from that, the scrambling of inter-and intra-DNA plane was achieved at the same time. In the next step, the process was done on DNA matrix which was permutated. The process consisted of XOR-operation of DNA utilizing the key matrix of DNA strands. The said key matrix was attained through the 2 one-dimensional chaotic system mapping. Lastly, confused DNA-matrix was decoded, which resulted in the encrypted image. A research [71] based on using the combination of chaotic maps with technique of differential encoding

was proposed as a method to perform the encryption of the digital images. For the confusion phase, the randomization of coordinate points in the image were performed by scrambling algorithm using 2D Arnold map. The value of pixels were changed in perspective to the original image using the magic square. The techniques of differential encoding focused on bit-level security, whereas chaotic maps focused on leveling up the complexity while increasing randomness in the diffusion phase.

In the research [72], the conjoined use of RNA codons and MSC (Magical Square Chaotic Algorithm) presented us with a hybrid method for encryption of plain images. The starting value used for chaotic function LS2 was generated by algorithm SHA-256. MSC was used for moving pixels of the image. Again, to move pixel values, chaotic function and RNA codons were employed. Lastly, algorithm for genetic operators were used to optimize the process. The proposed algorithm rendered nice security effects to resist varying future attacks. An other relevant work [73] aimed at improving the chaotic system. Indeed, improving a chaotic map would be beneficial for the potential algorithms of encryption. The reported work employed a 1D logistic map integrated with feedback control and a coupling mechanism to enhance the degree of randomness by dynamically altering the chaotic sequence for selective position shifting. The simulation of this method entailed that the result of 2D chaotic maps were improved in terms of complexity and bigger chaos range. Furthermore, the use of improved chaotic sequences used in correspondence to octree principle performed better with greater security.

In another study [74], the use of 3D Arnold transform, RSA and MPFrDCT (multiple parameter fractional discrete cosine transform) provided a unique technique to encrypt and decrypt the digital images. The color maps were removed in the starting phase in order to convert them in their indexed form. This from was taken as RGB channel. 3D Arnold Transform was employed to feed these RGB values, which was used to change the pixel values, performing substitution and permutation. For the last step, multiple parameter fractional discrete cosine transforms were used to change the given domain into the frequency domain of image that had been encrypted. This technique was dependent not only on the secret keys but also on how they are arranged. In this way, simply knowing the secret keys will do no harm to the cipher image. Moreover, the combination of magic squares with chaotic map and particle swarm optimization provided an other algorithm to perform encryption [75]. The values of magic square were filled with the breaking up of image into single 8-bit (1-byte )blocks. This initiated the particle swarm optimization (PSO) process, as the cipher image was used as particles in the system. Moreover, many works have produced novel image encryption algorithms [76–78]

## 3 Rössler chaotic system

The theory of chaos refers to the phenomenon that the tiniest change in the initial set of states, as well as system parameters, results in a huge change in future states of the system [23]. Using this philosophy of chaos, mathematicians and scientists have come up with a huge number of chaotic systems. These systems are highly dependent on system parameters and initial states. Besides, image cryptographers have used them abundantly in developing their image ciphers. Low dimensional chaotic systems do not render the much chaotic and random data. So, their employment in writing the image ciphers do not provide the nice results of different security parameters. However, the higher dimensional maps render much chaotic data, hence, they cause to produce relatively better results of security parameter. However, they cause computational time overhead for the ciphers. By considering these two extremes, the current study has

employed the 3D Rössler chaotic system [13]. In 1976, Otto E. Rössler proposed a system of three differential equations with one of them has non-linear term that has a dynamic behavior able to produce chaos. Following are the mathematical equations for the selected system.

$$\dot{x} = -y - z$$
$$\dot{y} = x + ay$$
$$\dot{z} = b + z(x - c)$$

Here $x$, $y$ and $z$ are the system states and $a$, $b$, $c$ correspond to the system parameters. The Rössler attractor achieves the chaotic system upon the values of $a = 0.2$, $b = 0.2$, and $c = 5.7$ (Figs 1 to 4). Apart from that, the Bifurcation diagram of the chaotic system can be seen in the Fig 5.

The Lyapunov exponent analysis was conducted by varying the control parameter $c$ revealing regions of chaotic behavior characterized by positive exponent values. This validates the system's sensitivity to initial conditions and confirms the presence of chaos, further supporting its suitability for secure image encryption (Fig 6).

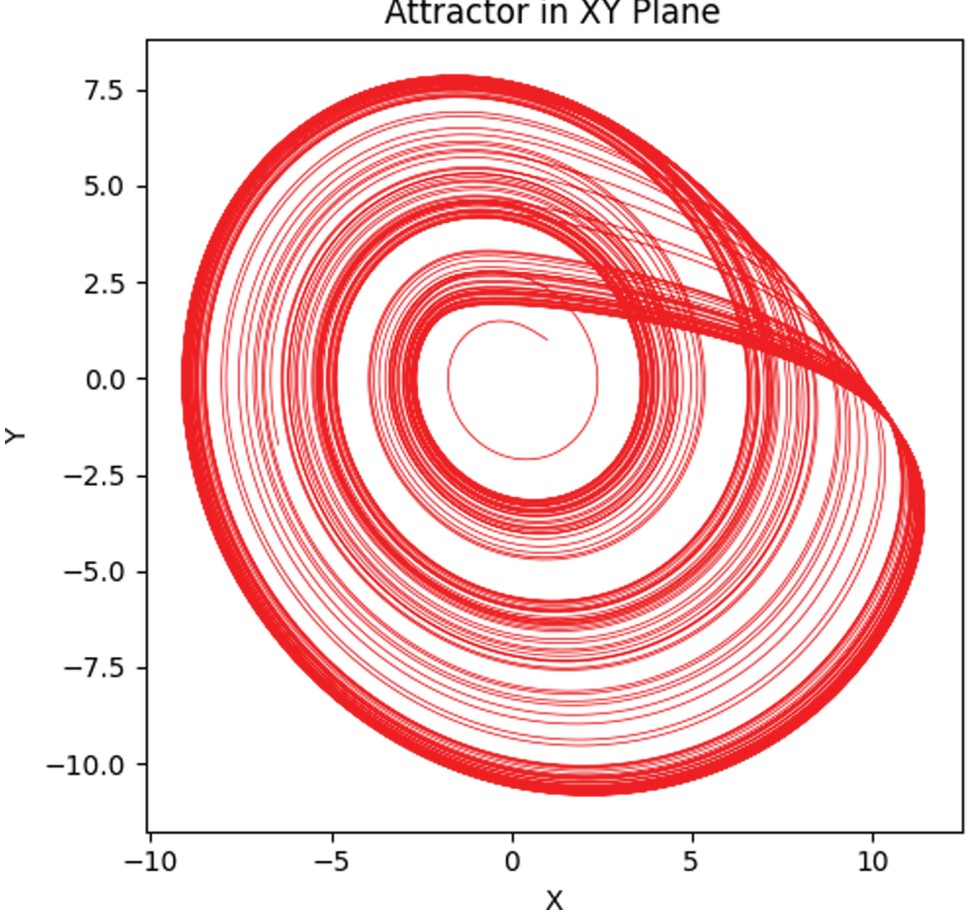

**Fig 1. The Rössler attractors in XY plane.**

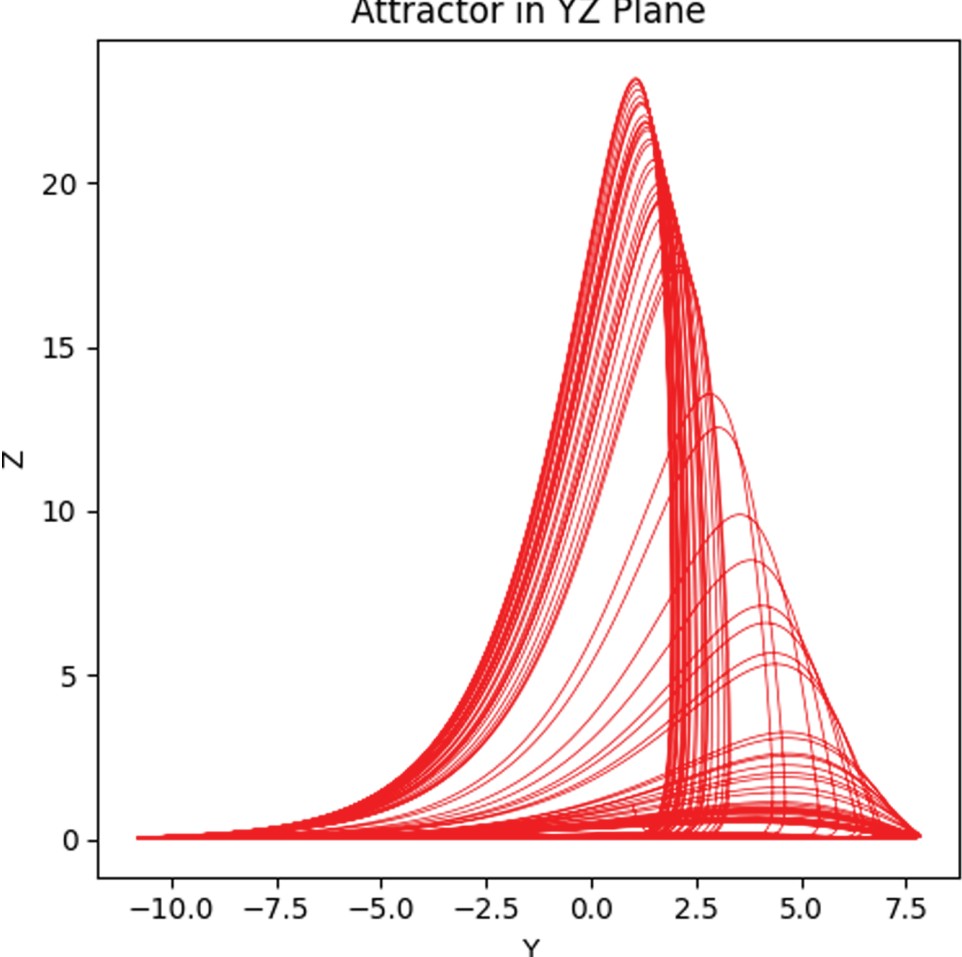

**Fig 2. The Rössler attractors in YZ plane.**

## 4 Materials and methods

The underlying idea of the current research endeavor is to safeguard the images from cyber security attacks as they travel through the open network like the Internet. Before transmitting these images, they are encrypted by using some encryption algorithm. Once, they are encrypted, they get transmitted. As these cipher images reach to their destination, they are converted to their original forms through the application of the decryption algorithm over them. Moreover, Rössler chaotic map has been employed to generate the streams of random numbers. These numbers facilitate in doing the diffusion and scrambling operations over the pixel data of these images.

### 4.1 Generation of key stream

Let the input plain image is *img* whose size is $m \times n$. Plaintext sensitivity is an important feature to thwart hidden threats of differential attack. This feature has been embedded by taking the pixels' sum of given input image. This sum has been, in turn, employed to temper the initial values of the chaotic system being employed in this work. Suppose that *sum* is sum of

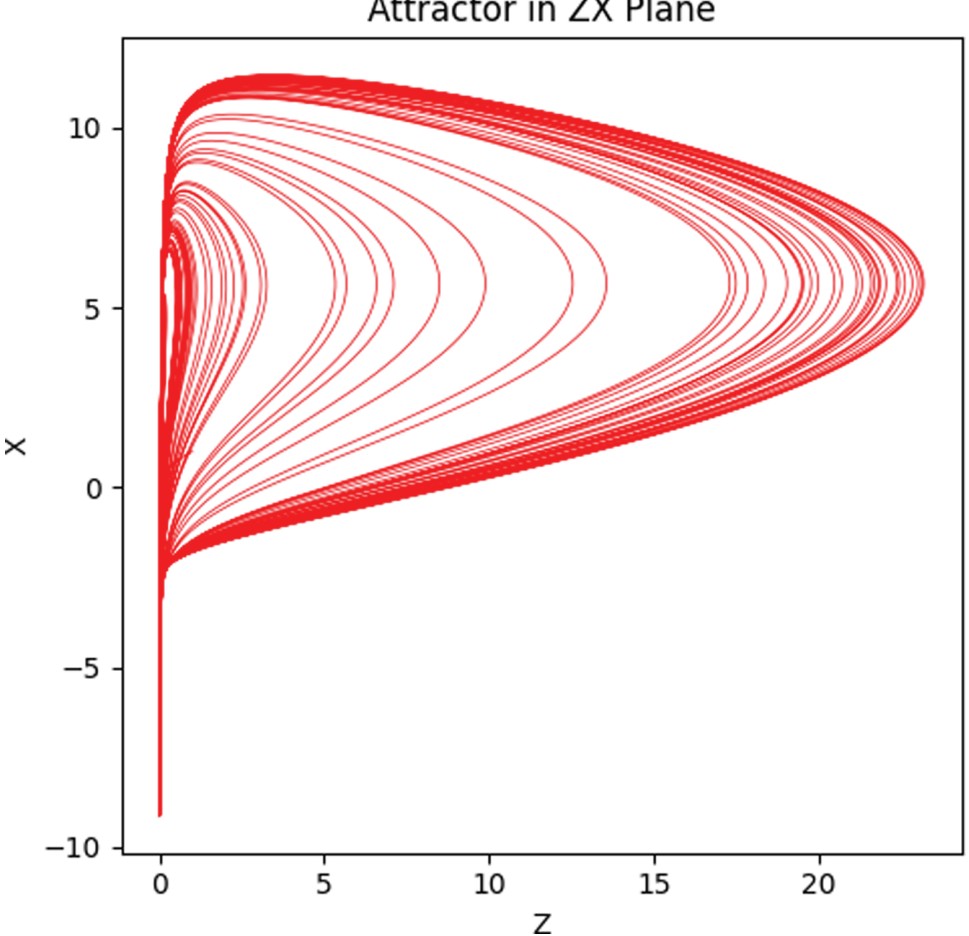

**Fig 3. The Rössler attractors in ZX plane.**

pixels' intensity values of given input image. Assign 1.90 to $x_0$ and update its value according to the following equation:

$$x_0 = 1.90 + \frac{sum}{2^{30}} \tag{1}$$

Assign the other values to the primary states and system parameters of chaotic system as $y_0 = 0.6878, z_0 = 0.3741, a = 0.2, b = 0.2, c = 5.7$. Now the Rössler chaotic map has been ignited by giving the updated secret key. This map rendered three streams of random numbers, i.e., $\{x_t\}_{t=1}^{mn}, \{y_t\}_{t=1}^{mn}$ and $\{z_t\}_{t=1}^{mn}$. These random numbers are very raw. They must to tailored to fit to the algorithmic logic of pixels scrambling, this study has conceived. So pass them through Equation 2 with these parameters $\{x_t\}_{t=1}^{mn}, \{y_t\}_{t=1}^{mn}, \{z_t\}_{t=1}^{mn}$, $m$ and $n$. It is to be noted that $(m, n)$ corresponds to the size of the given image. This equation provides the three new streams of random numbers, i.e., $\{hori_t\}_{t=1}^{mn}, \{verti_t\}_{t=1}^{mn}$ and $\{mask_t\}_{t=1}^{mn}$ with the corresponding ranges of $[1, 2, 3, ..., m], [1, 2, 3, ..., n]$ and $[0, 1, 2, ..., 255]$.

$$\begin{bmatrix} hori(t) \\ verti(t) \\ mask(t) \end{bmatrix} = \begin{bmatrix} \mathrm{mod}(\lfloor u(t) \cdot 10^{14} \rfloor, m) + 1 \\ \mathrm{mod}(\lfloor v(t) \cdot 10^{14} \rfloor, n) + 1 \\ \mathrm{mod}(\lfloor u(t) \cdot 10^{14} \rfloor, 256) \end{bmatrix} \tag{2}$$

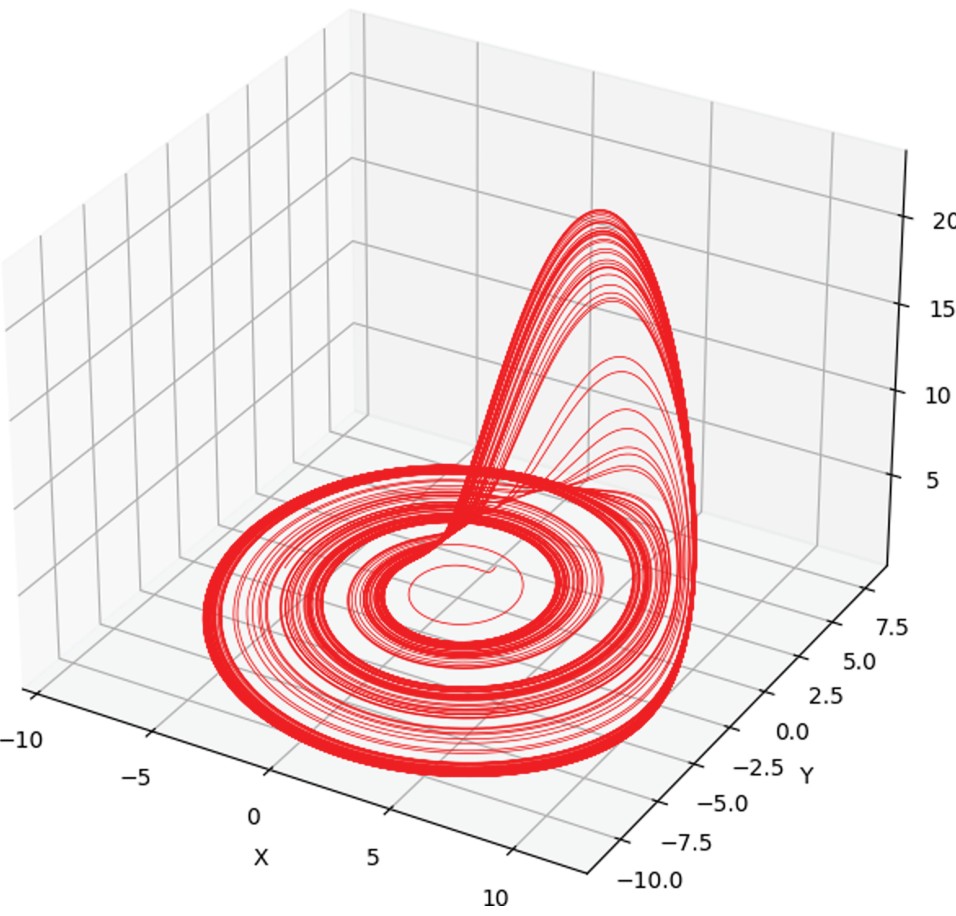

**Fig 4. The Rössler attractors in 3D.**

## 4.2 Proposed image encryption scheme

The image scrambler consists of two algorithms, i.e., Row based scrambling and Column based scrambling. Since, Row-Based and Column-Based scramblings are symmetric, so we have drawn only the schematic diagram of how the row based scrambling has been carried out (see Fig 7). Call the Algorithm 1 for the scrambling of the given image *img* with the parameters of *img* and *hori*.

Here we explain the steps of the Algorithm 1.

Line 1 finds the size of the given image *img* and assign it to the $[m, n]$. Line 2 initializes a variable called *pointer* with the value of 1. A *goto* control structure has been employed in this algorithm. Line 3 corresponds to the label *start* of this structure. The *goto* statement is fired at the line 5 and the control shifts to the label *end* (line 37) if the condition at the line 4 gets true. Lines (7-11) create a scrambled image S1 (with the size of $m \times n$) each of whose cells are occupied with the value of -1. Lines (12-15) initialize the two arrays named as *flagSI* and *flagPI* with the value -1. These values will act as a filter to make the relevant decisions. In the

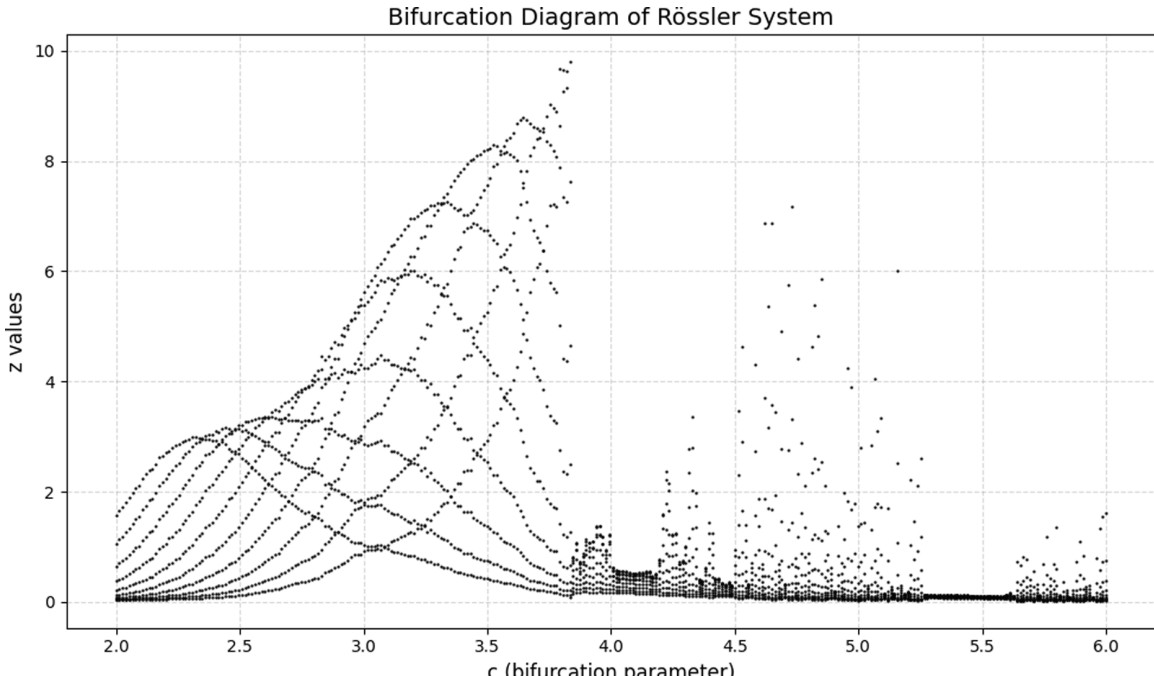

**Fig 5. Bifurcation diagram of the Rössler chaotic system.**

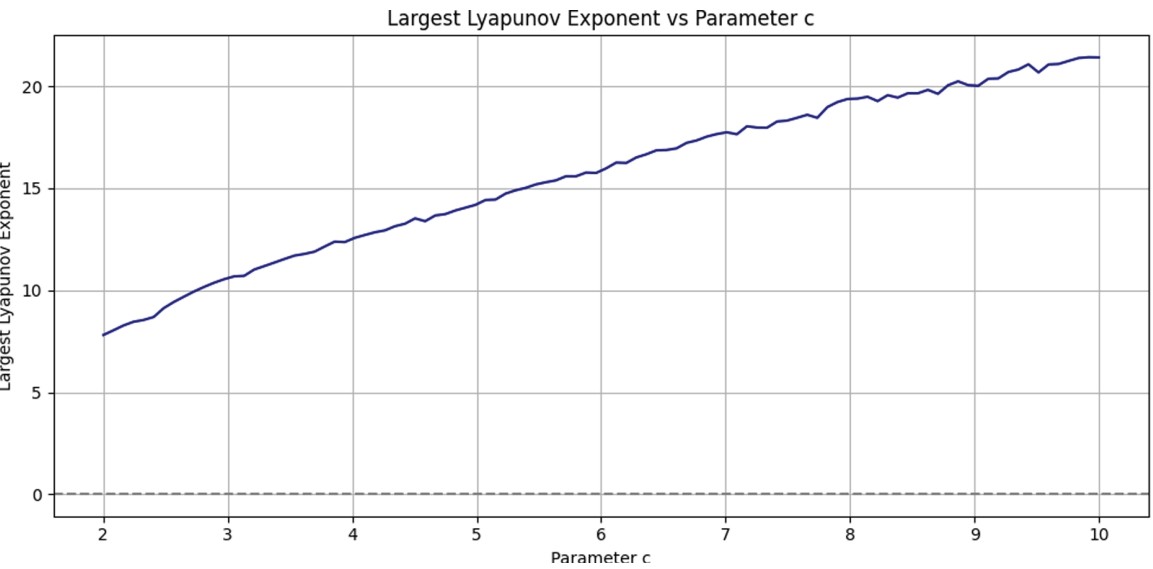

**Fig 6. Lyapunov Exponent of the Rössler chaotic system.**

line 16, a slice of random numbers $[1, 2, 3, ..., m]$ has been taken in the array *horiTemp* from the array *hori*. These random numbers will facilitate in carrying out the confusion operation for encryption. Lines (17-23) assign the rows from the plain image *img* to the scrambled image $S1$. Line 18 checks whether the $horiTemp(k)^{th}$ index is empty? If it is, the line 19 assigns the $k^{th}$ row from the plain image *img* to the $horiTemp(k)^{th}$ row of the scrambled

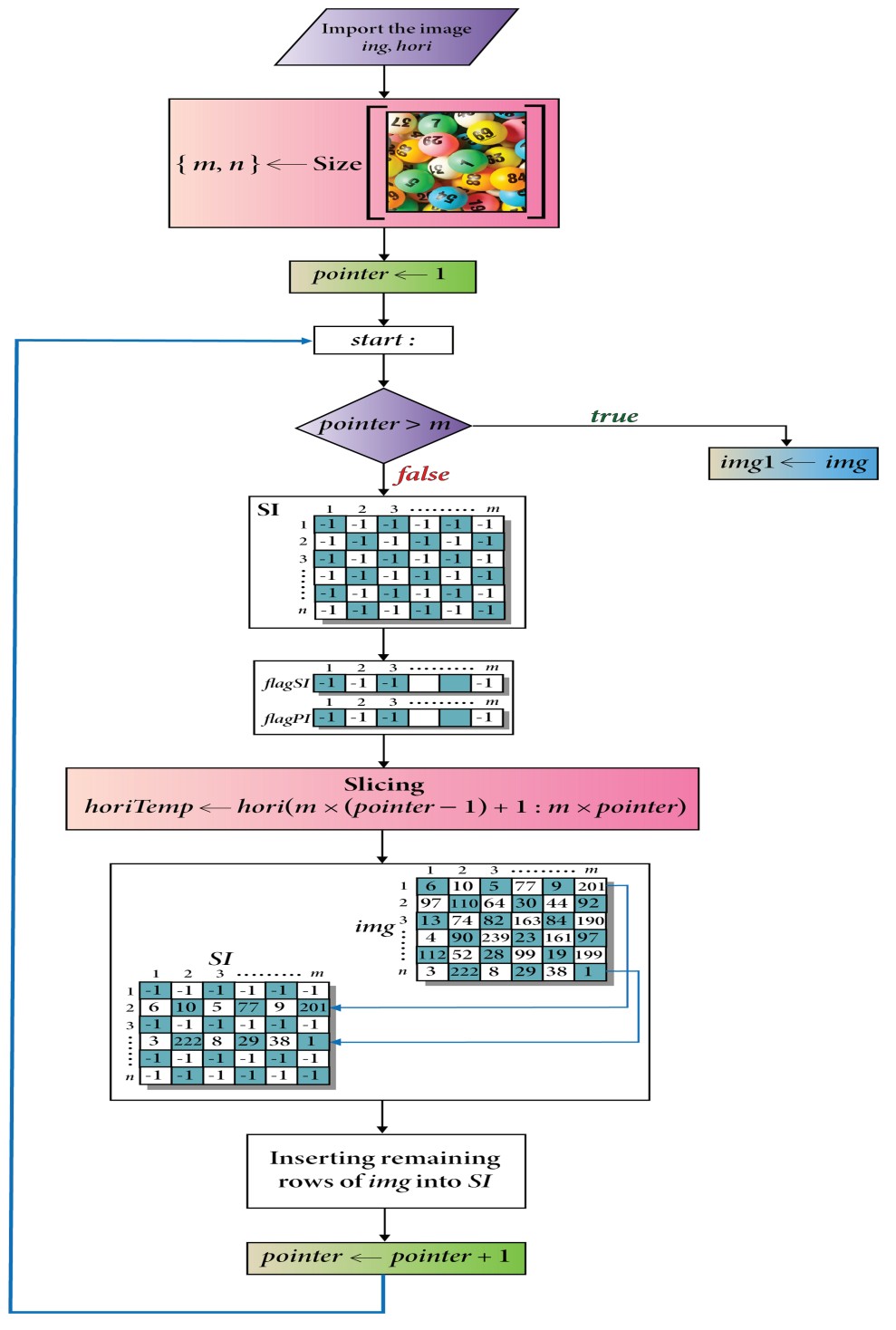

**Fig 7. Row based image scrambler.**

image. Lines 20 and 21 assign 0 to the $k^{th}$ index of the arrays *flagPI* and *flagSI*. Lastly, the lines (24–33) assign the remaining rows from the plain image to the scrambled image *SI*. In order to repeat this process, line 34 updates the image *img* with the array of *S1*. Line 35 increases

**Algorithm 1.** Row based scrambling

**Input**: *img*, *hori*
**Output**: *img*1
1: $[m,n] \leftarrow size(img)$
2: $pointer \leftarrow 1$
3: $start$:
4: **if** $pointer > m$ **then**
5: $goto\ end$
6: **end if**
7: **for** $i \leftarrow 1$ to $m$ **do**
8: **for** $j \leftarrow 1$ to $n$ **do**
9: $S1(i,j) \leftarrow -1$
10: **end for**
11: **end for**
12: **for** $row \leftarrow 1$ **to** $m$ **do**
13: $flagSI(row) \leftarrow -1$
14: $flagPI(row) \leftarrow -1$
15: **end for**
16: $horiTemp \leftarrow hori(m \times (pointer - 1) + 1 : m \times pointer)$
17: **for** $k \leftarrow 1$ **to** $m$ **do**
18: **if** $flagSI(horiTemp(k)) = -1$ **then**
19: $SI(horiTemp(k),:) \leftarrow img(k,:)$
20: $flagPI(k) \leftarrow 0$
21: $flagSI(k) \leftarrow 0$
22: **end if**
23: **end for**
24: $index \leftarrow 0$
25: **for** $i \leftarrow 1$ **to** $m$ **do**
26: **if** $flagSI(i) = 0$ **then**
27: **while** $flagPI(index + 1) = 0$ **do**
28: $index \leftarrow index + 1$
29: **end while**
30: $S1(i,:) \leftarrow img(index,:)$
31: $index \leftarrow index + 1$
32: **end if**
33: **end for**
34: $img \leftarrow SI$
35: $pointer \leftarrow pointer + 1$
36: $goto\ start$
37: $end$:
38: $img1 \leftarrow img$

the value of variable *pointer* by the value of 1. Line 36 shifts the control to the label *start* to repeat the process. As soon as the condition at line 4 evaluates to be *true*, control is shifted to the label *end*. Lastly, the line 38 assigns the scrambled image *img* to the variable *img*1.

In the same fashion, to introduce an added layer of security, call the Algorithm 2 for the scrambling of the given image *img*1 with the parameters of *img*1 and *verti*. It is to be noted

**Algorithm 2.** Column based scrambling

**Input:** $img1$, $verti$
**Output:** $img2$
1: $[m, n] \leftarrow size(img1)$
2: $pointer \leftarrow 1$
3: $start$:
4: **if** $pointer > n$ **then**
5: $goto\ end$
6: **end if**
7: **for** $i \leftarrow 1$ **to** $m$ **do**
8: **for** $j \leftarrow 1$ **to** $n$ **do**
9: $S2(i, j) \leftarrow -1$
10: **end for**
11: **end for**
12: **for** $column \leftarrow 1$ **to** $n$ **do**
13: $flagSI \leftarrow -1$
14: $flagPI \leftarrow -1$
15: **end for**
16: $vertiTemp \leftarrow verti(n \times (pointer - 1) + 1 : n \times pointer)$
17: **for** $k \leftarrow 1$ **to** $m$ **do**
18: **if** $flagSI(vertiTemp(k)) = -1$ **then**
19: $S2(:, vertiTemp(k)) \leftarrow img1(:, k)$
20: $flagPI \leftarrow 0$
21: $flagSI \leftarrow 0$
22: **end if**
23: **end for**
24: $index \leftarrow 0$
25: **for** $i \leftarrow 1$ **to** $m$ **do**
26: **if** $flagSI(i) = 0$ **then**
27: **while** $flagPI(index + 1) = 0$ **do**
28: $index \leftarrow index + 1$
29: **end while**
30: $S2(:, i) \leftarrow img1(:, index)$
31: $index \leftarrow index + 1$
32: **end if**
33: **end for**
34: $img1 \leftarrow S2$
35: $pointer \leftarrow pointer + 1$
36: $goto\ start$
37: $end$:
38: $img2 \leftarrow img1$

that the same operations have been carried out in this algorithm but based on the assignment of columns from the image $img1$ to the randomly chosen indices for the columns of the scrambled image. Lastly, this algorithm returns the image $img2$.

As the given image has been sufficiently scrambled both row- and column-wise, now to embed the diffusion effects in the scrambled image $img2$, reshape the scrambled image $img2$ to the size of $1 \times mn$. Carry out the following operation between the scrambled image $img2$ and the key image $mask$.

$$img3(i) = img2(i) \oplus mask(i) \tag{3}$$

For $i = 1, 2, 3, ..., mn$. Resize the image $img3$ to $m \times n$. $img3$ is the final cipher image carrying both the effects of confusion and diffusion.

It is to be noted that the proposed cipher was developed using the tenets of private key cryptography, so its steps would be an exact inverse of the steps of the encryption algorithm.

## 5 Machine simulation

In this section, the suggested encryption algorithm for digital images would be demonstrated by taking some sample images. For this purpose, we have taken images of Chair, Bride, Flowers, and Hail. The size of all these images is $256 \times 256$. For simulation purposes, MATLAB 2024 has been used. Moreover, the precision used was 64-bit according to the IEEE Standards. Figs 8 to 11 show the plain grayscale images. Moreover, Figs 12 to 15 and Figs 16 to 19 show their encrypted and decrypted versions. One can notice that plain images got completely turned into a blurry and cloudy format which cannot be recognized at all. This signals the fact that the suggested image encryption algorithm is potent enough to thwart hackers. Besides, the Figs 20 to 25 show the plain, cipher and decrypted images whose size is $512 \times 512$.

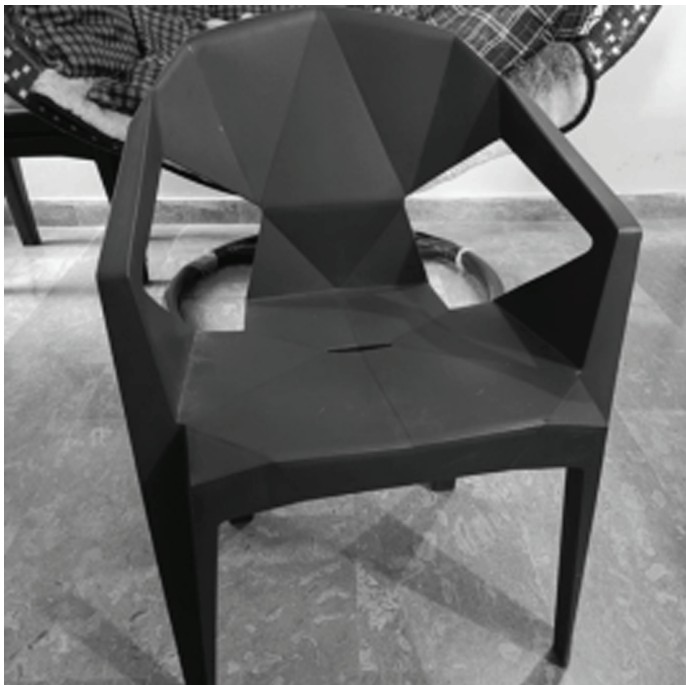

**Fig 8. Chair ($256 \times 256$) plain image.**

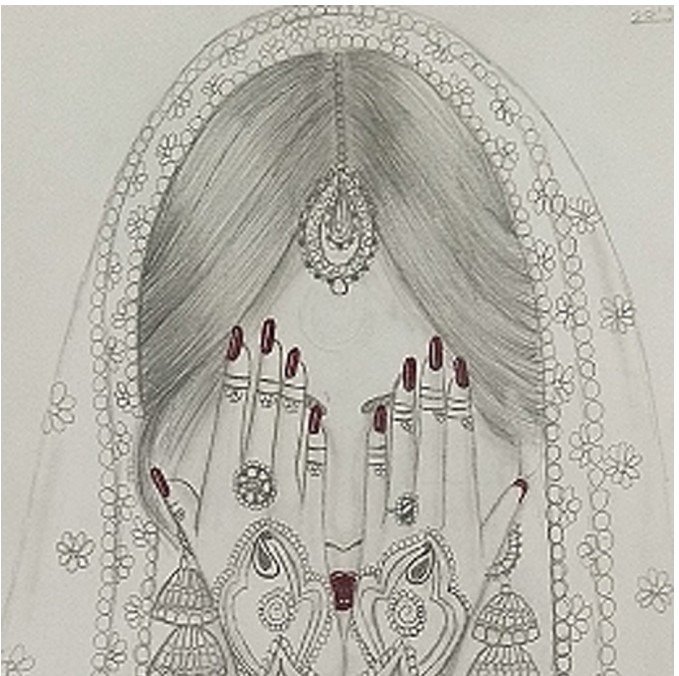

**Fig 9. Bride (256 × 256) plain image.**

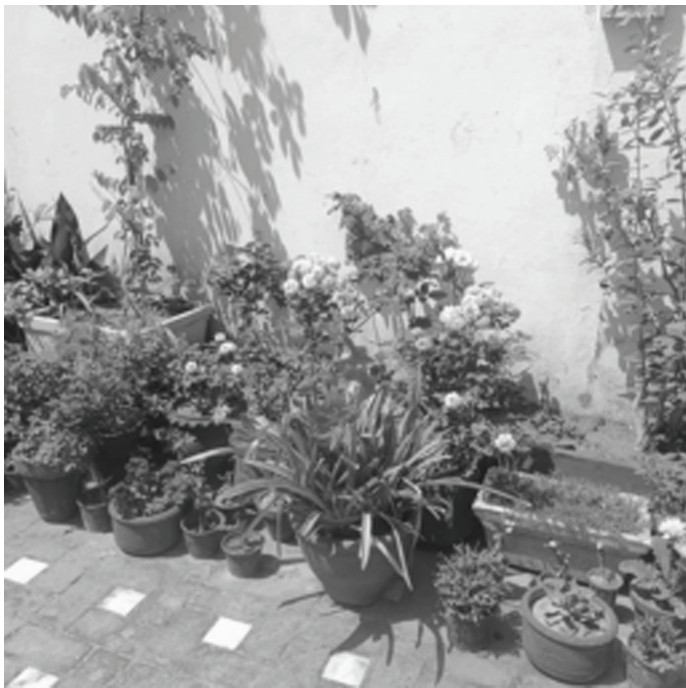

**Fig 10. Flower (256 × 256) plain image.**

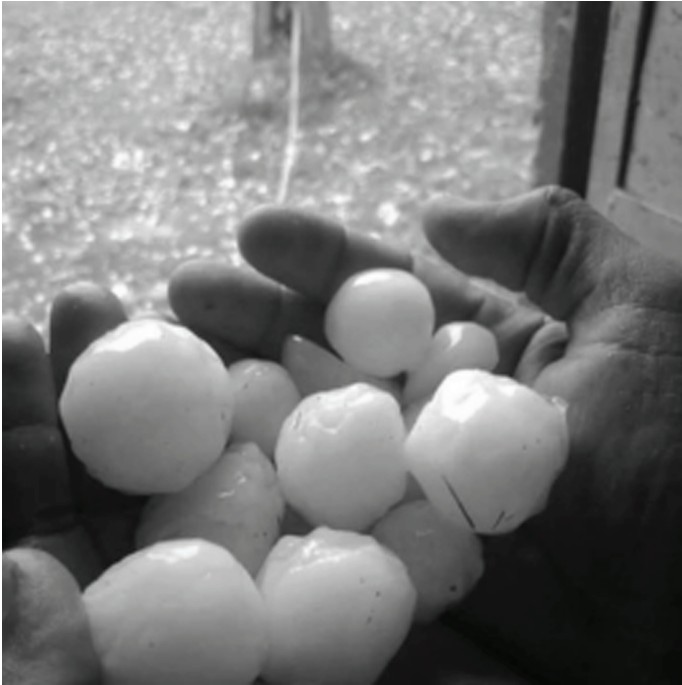

**Fig 11. Hailstones (256 × 256) plain image.**

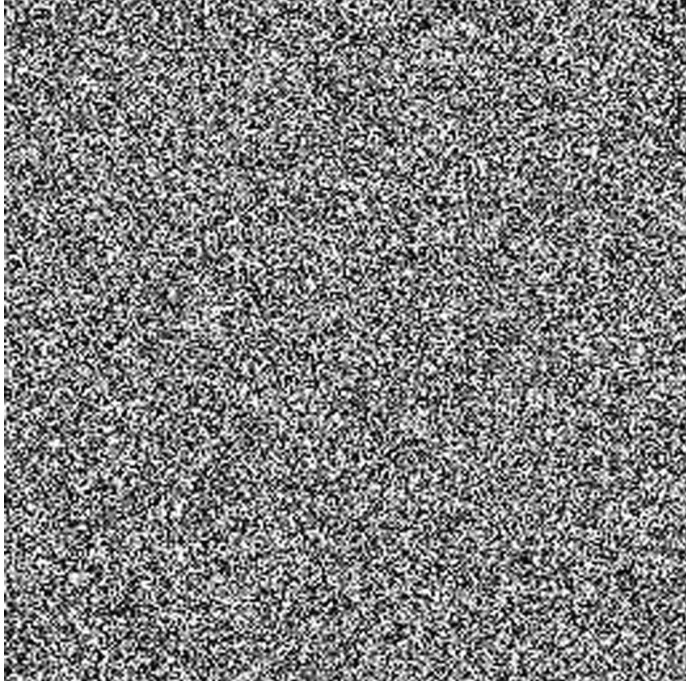

**Fig 12. Chair (256 × 256) plain image.**

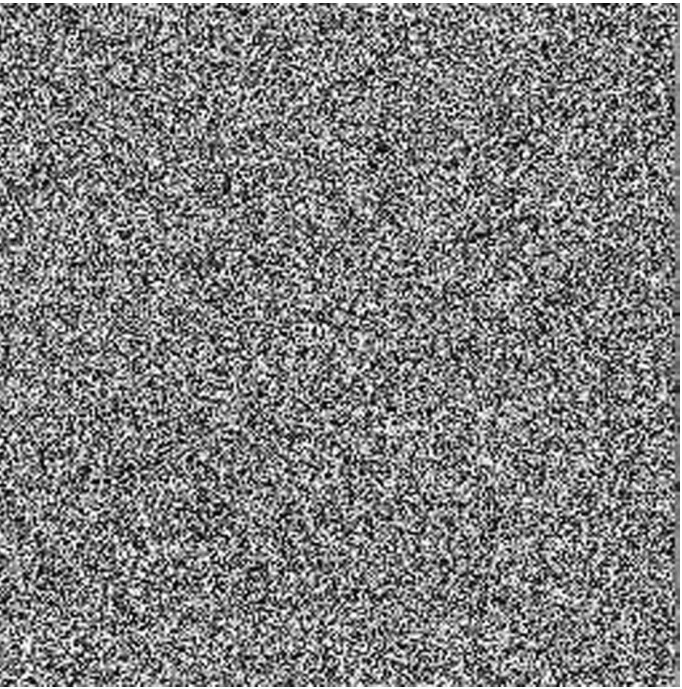

**Fig 13. Bride (256 × 256) cipher image.**

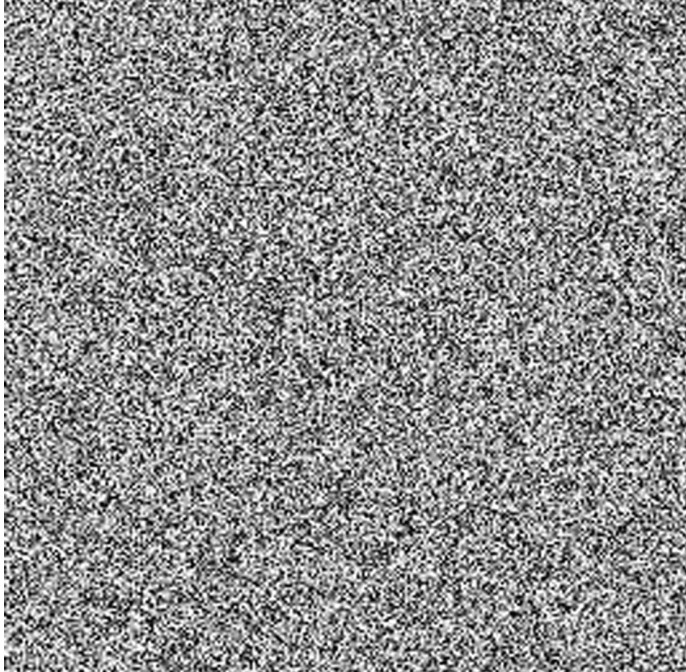

**Fig 14. Flower (256 × 256) cipher image.**

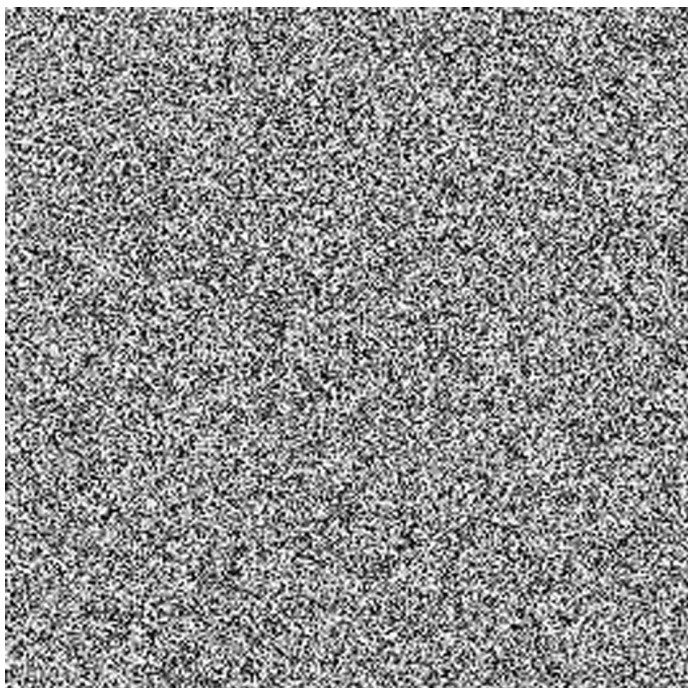

**Fig 15. Hailstones (256 × 256) cipher image.**

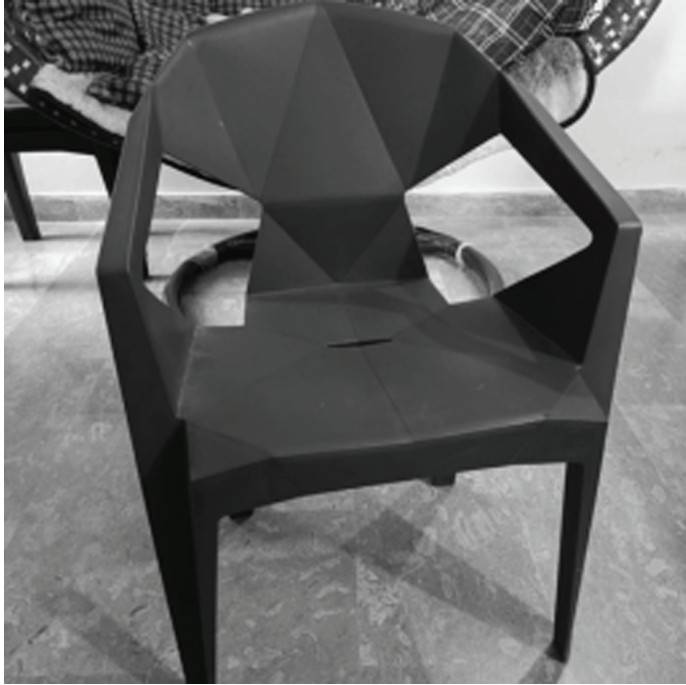

**Fig 16. Chair (256 × 256) decrypted image.**

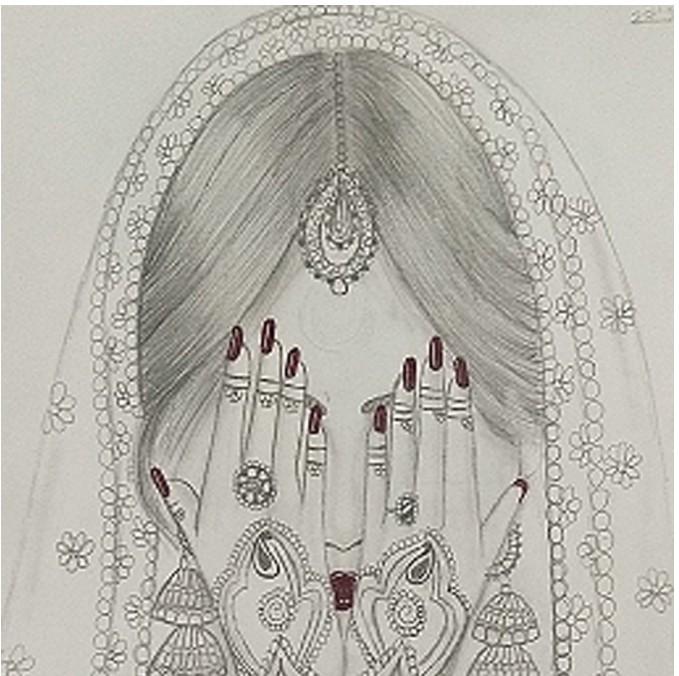

**Fig 17. Bride (256 × 256) decrypted image.**

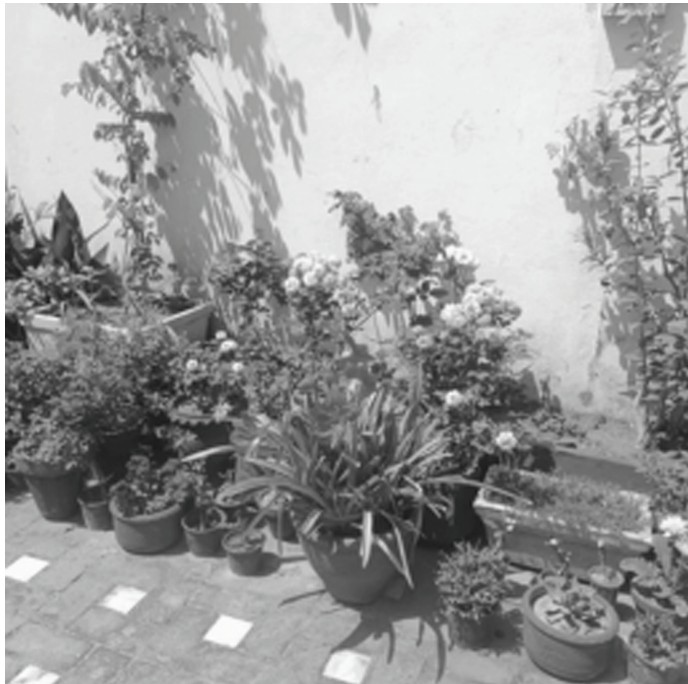

**Fig 18. Flower (256 × 256) decrypted image.**

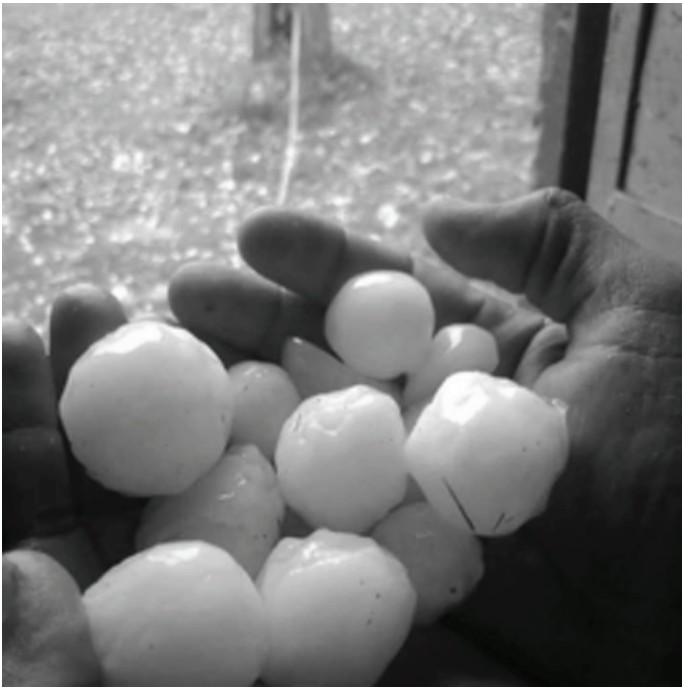

**Fig 19. Hailstones (256 × 256) decrypted image.**

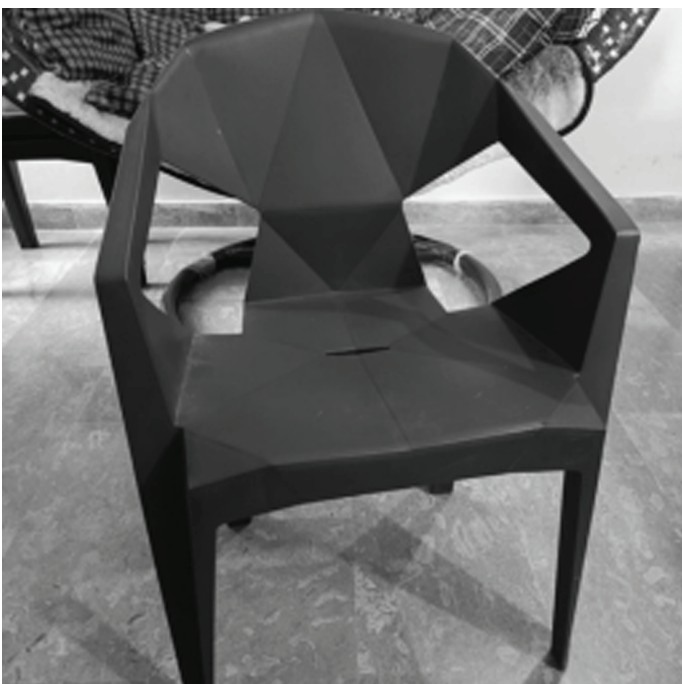

**Fig 20. Chair (512 × 512) plain image.**

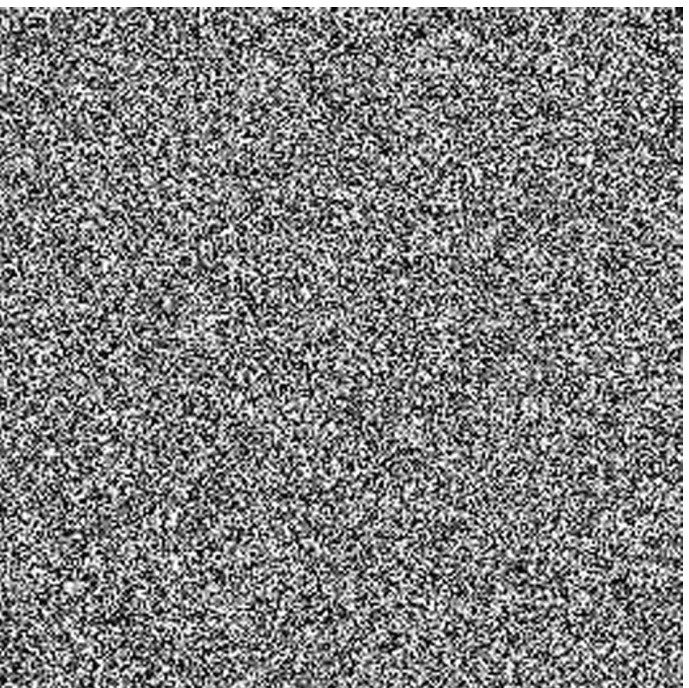

**Fig 21. Chair (512 × 512) cipher image.**

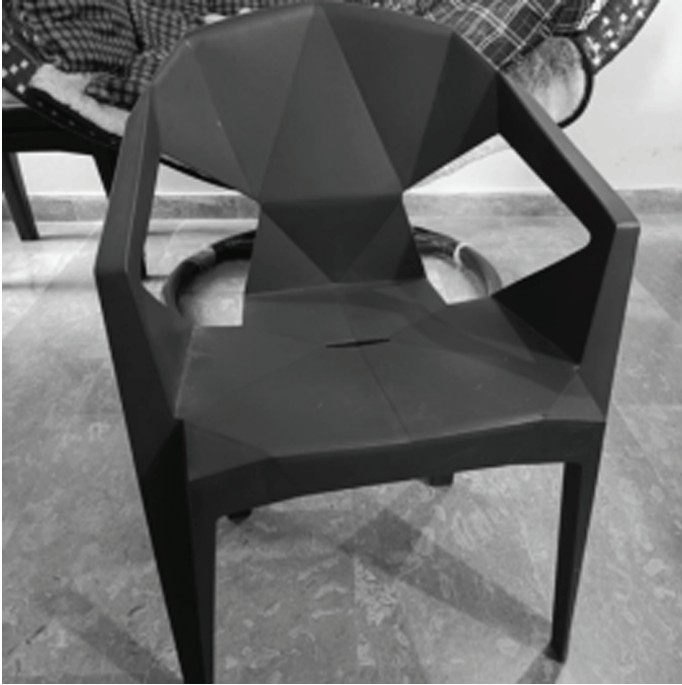

**Fig 22. Chair (512 × 512) decrypted image.**

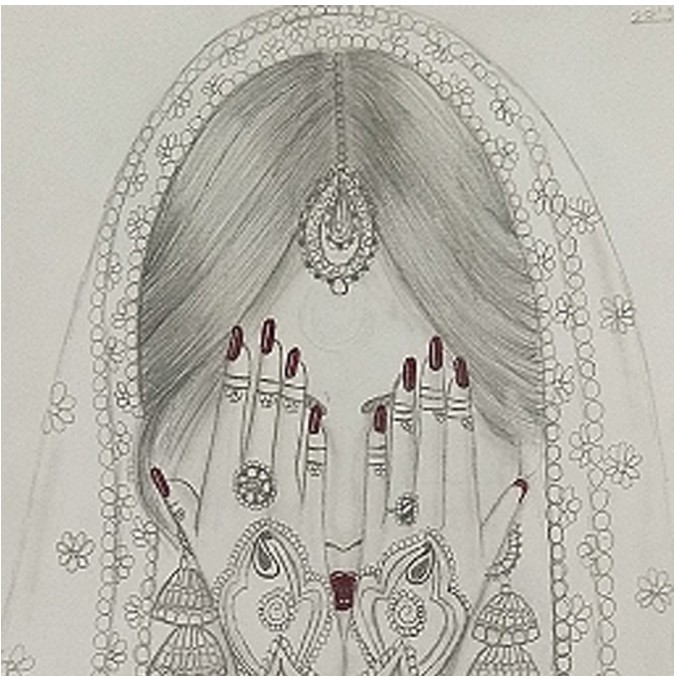

**Fig 23. Bride (512 × 512) plain image.**

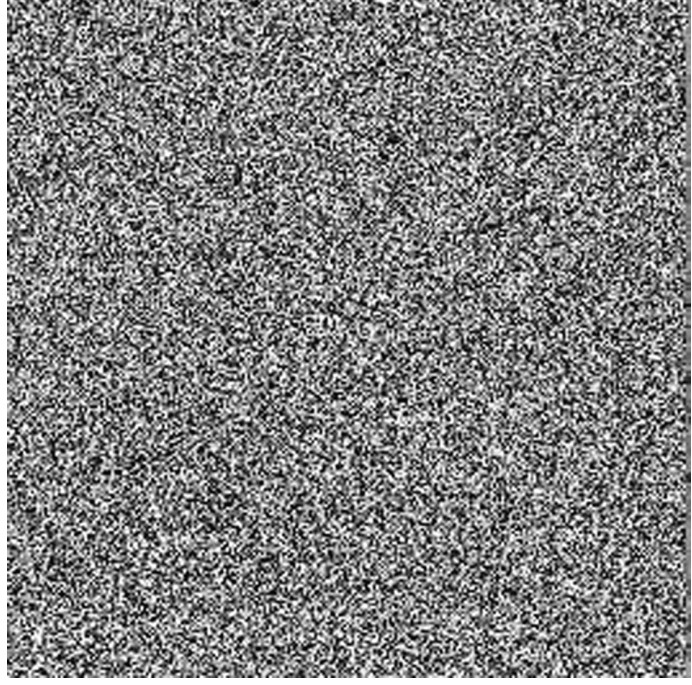

**Fig 24. Bride (512 × 512) cipher image.**

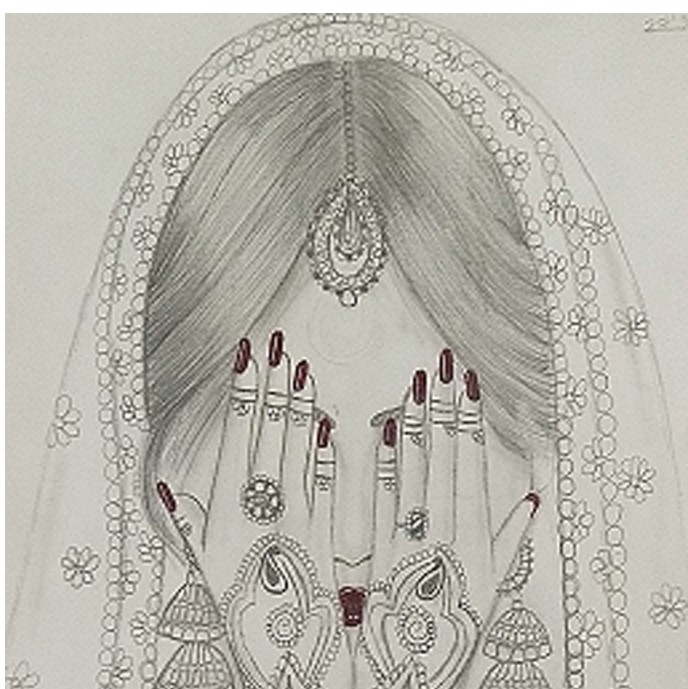

**Fig 25. Bride (512 × 512) decrypted image.**

## 6 Performance and security analysis of image cipher

In this section, the suggested image cipher would be subjected to some chosen security parameters to demonstrate their immunity against various attacks.

### 6.1 Key space

A key space represents the complete set of all possible keys within an encryption system. It typically includes all initial values and system parameters essential to the encryption scheme. A sufficiently large key space is crucial for resisting brute-force attacks, where an adversary systematically tests all possible keys to decrypt the data. Cryptographers recommend a minimum key space size of $2^{100}$ [40] to ensure security. A limited key space makes an encryption system vulnerable to such attacks, compromising its robustness.

In the proposed encryption algorithm, key parameters—including $x_0, y_0, z_0, a, b$, and $c$—are derived from a chaotic map. Given a computational precision of $10^{-15}$, the resulting key space is approximately $10^{90}$, significantly exceeding the minimum recommended threshold of $2^{100}$. This ensures strong resistance against brute-force attacks, reinforcing the security of the proposed image cipher.

Furthermore, Table 1 presents a comparative analysis of the key space of our scheme against previously published studies. Unfortunately, our study couldn't beat any cited work vis-à-vis the security parameter but even then we contend that the suggested work is safe from the potential brute force attack since $2^{298} \gg 2^{100}$.

### 6.2 Key sensitivity

A robust cipher must exhibit high sensitivity to its secret key, meaning even the slightest alteration in any key parameter should produce a completely different result. This section

**Table 1. Suggested algorithm's key space along with comparison.**

| Algorithm | Key space |
|---|---|
| Suggested | $10^{90} \approx 2^{298}$ |
| Ref. [44] | $10^{195}$ |
| Ref. [80] | - |
| Ref. [79] | $10^{144}$ |
| Ref. [81] | $2^{430}$ |

demonstrates the key sensitivity of the proposed image cryptosystem. The approach is as follows: first, a cipher image is generated using the encryption algorithm with a given secret key. Next, the encryption process is repeated with an almost identical key, except for a minute modification in one of its parameters. The resulting changes in pixel values are then analyzed. If the difference is substantial, it confirms the cipher's strong key sensitivity. For this experiment, the secret key is defined as $\{x_0, y_0, z_0, a, b, c\}$ (say $Key_0$). Using this key, the Chair image (Fig 8) is encrypted. Subsequently, a minuscule perturbation of $10^{-14}$ is introduced in one of the key variables, specifically $x_0$. To put in other words, let $x_0' = x_0 + 10^{-14}$. Thus, the modified key set obtained is $\{x_0', y_0, z_0, a, b, c\}$ denoted as $Key_1$. The Chair image from Fig 8 has been encrypted using the modified key $Key_1$. The resulting pixel variations are presented in Table 2. The difference rates between the cipher images generated with the original key $Key_0$ and the altered keys $Key_k$ ($k = 1, 2, \ldots, 12$) have been computed. The average difference rate is 99.62%, surpassing the results reported in previous studies [62,63]. This demonstrates that the proposed cipher exhibits higher sensitivity to the secret key.

## 6.3 Statistical randomness analysis

Cipher images must be resilient to statistical attacks to ensure security. The NIST randomness test is a crucial measure for evaluating the randomness of encrypted data [1]. For the encrypted bit stream to be considered random, the $p$–$value$ must be greater than 0.01 for the specified input parameters. Table 3 presents the NIST randomness test results [2] for the Chair cipher image. As observed, the $p$–$value$ exceeds the 0.01 threshold across all sixteen parameters, confirming that the pixels in the encrypted Chair image exhibit a high degree of randomness.

**Table 2. Rates of difference between two images produced by minutely distinct keys.**

| Secret security keys | Difference rates(%) | | | |
|---|---|---|---|---|
| | Chair | Bride | Flowers | Hail |
| $Key_1 (x_0' = x_0 + 10^{-14})$ | 99.5965 | 99.6183 | 99.6192 | 99.5996 |
| $Key_2 (y_0' = y_0 + 10^{-14})$ | 99.6098 | 99.5933 | 99.6287 | 99.5941 |
| $Key_3 (z_0' = z_0 + 10^{-14})$ | 99.6276 | 99.5712 | 99.6896 | 99.5612 |
| $Key_4 (a' = a + 10^{-14})$ | 99.6587 | 99.5977 | 99.6765 | 99.5623 |
| $Key_5 (b' = b + 10^{-14})$ | 99.6447 | 99.6098 | 99.6587 | 99.6087 |
| $Key_6 (c' = c + 10^{-14})$ | 99.6135 | 99.5351 | 99.6322 | 99.5723 |
| $Key_7 (x_0' = x_0 - 10^{-14})$ | 99.6176 | 99.5957 | 99.6387 | 99.5766 |
| $Key_8 (y_0' = y_0 - 10^{-14})$ | 99.6046 | 99.5987 | 99.6295 | 99.5947 |
| $Key_9 (z_0' = z_0 - 10^{-14})$ | 99.6277 | 99.5753 | 99.6671 | 99.6638 |
| $Key_{10} (a' = a - 10^{-14})$ | 99.6587 | 99.5977 | 99.6765 | 99.6308 |
| $Key_{11} (b' = b - 10^{-14})$ | 99.6411 | 99.6098 | 99.6598 | 99.6487 |
| $Key_{12} (c' = c - 10^{-14})$ | 99.6134 | 99.5957 | 99.6306 | 99.6533 |
| **Average** | **99.62** | **99.59** | **99.65** | **99.61** |
| **Average of all** | **99.62** | - | - | |

**Table 3. Statistical randomness test results.**

| Test # | Test Name | p–value | Result |
|---|---|---|---|
| 1 | Frequency (Monobit) | 0.2176 | Success |
| 2 | Block Frequency ($m = 128$) | 0.3498 | Success |
| 3 | Cumulative Sums (Forward) | 0.2299 | Success |
| 4 | Cumulative Sums (Reverse) | 0.3098 | Success |
| 5 | The Run Test | 0.1298 | Success |
| 6 | Longest Run of Ones | 0.2982 | Success |
| 7 | Rank | 0.1765 | Success |
| 8 | DFT Spectral | 0.2198 | Success |
| 9 | Non Overlapping Template ($m = 9$, $B = 000000001$) | 0.0689 | Success |
| 10 | Overlapping Template ($m = 9$) | 0.2098 | Success |
| 11 | Universal Statistical Test | 0.1987 | Success |
| 12 | Approximate Entropy ($m = 10$) | 0.0387 | Success |
| 13 | Random Excursions | 0.4434 | Success |
| 14 | Random Excursions Variant | 0.3187 | Success |
| 15 | Serial ($m = 16$) | 0.4347 | Success |
| 16 | Linear Complexity ($M = 500$) | 0.2223 | Success |

## 6.4 Irregular deviation and maximum deviation security analyses

Cryptanalysts employ various techniques to break ciphers, one of which involves analyzing Irregular Deviation (ID) and Maximum Deviation (MD) in encrypted images. As their names suggest, these metrics quantify the degree of variation in pixel values between the plaintext and cipher images.

The mathematical formulations for these concepts are provided in [38].

$$IrregDevi = \sum_{j=0}^{n-1} |Devi_j - Amp| \tag{4}$$

and [38]

$$MaxDevi = \frac{Devi_0 + Devi_{n-1}}{2} + \sum_{j=1}^{n-2} Devi_j \tag{5}$$

According to the equations above, $Devi_j$ represents the amplitude of discrepancy between the histograms of the encrypted and plaintext images for a given index $j$. The parameter $Amp$ denotes the average sum of histogram values, while $n$ represents the total number of pixels.

For optimal security, the values of $IrregDevi$ should be smaller, indicating better uniformity, whereas $MaxDevi$ should be larger, reflecting stronger deviation. As shown in Table 4, the lower values of $IrregDevi$ confirm that the proposed image cipher maintains good pixel uniformity. Additionally, the average value of 39,384 for the selected images is an improvement over 43,506 reported in [38]. Similarly, the average $MaxDevi$ value of 66,402 in the proposed cryptosystem outperforms the previous benchmark of 58,691 [38].

## 6.5 Energy and contrast analyses

Contrast analysis measures the variability in pixel intensity values within a given image. Naturally, greater intensity variations indicate a more heterogeneous image. In other words, higher values of this security parameter suggest that the encrypted image contains a broader

**Table 4. Results of irregular and maximum deviation.**

| Image ciphers | Images | Irregular deviation | Maximum deviation |
|---|---|---|---|
| Our algorithm | Chair | 39,354 | 66,160 |
| | Bride | 41,287 | 66,145 |
| | Flowers | 40,563 | 67,093 |
| | Hail | 36,334 | 66,212 |
| | **Average** | **39,384** | **66,402** |
| Ref. [38] | Pepper | 43,506 | 58,691 |

range of gray levels, which is indicative of stronger security. This contrast measure is typically expressed as [38].

$$C = \sum_{u,v} |u - v|^2 \times z(u, v) \tag{6}$$

where $u$ and $v$ are the intensity values in the given image. Besides, $z(u, v)$ refers to gray-level co-occurrence matrices' numbers (*GLCM*).

This metric determines the frequency with which a pixel of grayscale value $u$ appears in a spatial relationship with a pixel of value $v$. The results for this crucial security measure, obtained from the given images, are summarized in Table 5.

Specifically, the average contrast value is 10.3433, which surpasses 8.6448 reported in [38] but is slightly lower than 10.5325 from [4]. These results demonstrate that the proposed image cipher achieves competitive security performance.

Apart from that, energy of an image corresponds to the sum of squared elements in the gray level co-occurrence matrix [38]

$$E = \sum_{u,v} z(u, v)^2 \tag{7}$$

In this equation, $z(u, v)$ represents the values in the gray-level co-occurrence matrix, as previously described. Notably, smaller values of this metric indicate stronger security effects. Table 6 presents the energy analysis results for both plaintext and ciphertext images. As expected, plaintext images exhibit higher energy values, whereas encrypted images show lower values, reinforcing the effectiveness of the encryption. Furthermore, the average energy value is 0.0156, which is better than 0.1650 reported in [38] and matches 0.0156 from [4].

## 6.6 Homogeneity analysis

Homogeneity ($H$) analysis measures the closeness of pixel intensity distribution within the Gray-Level Co-Occurrence Matrix (GLCM). It evaluates the uniformity of pixel brightness

**Table 5. Results of contrast analysis.**

| Image ciphers | Images | Plaintext | Ciphertext |
|---|---|---|---|
| Our algorithm | Chair | 0.6534 | 10.7612 |
| | Bride | 0.6231 | 10.2609 |
| | Flowers | 0.6124 | 10.2387 |
| | Hail | 0.6347 | 10.1126 |
| | **Average** | **0.6309** | **10.3433** |
| Ref. [38] | Pepper | | 8.6448 |
| Ref. [4] | Lena | | 10.5325 |

**Table 6. Results of energy analysis.**

| Image ciphers | Images | Plaintext | Ciphertext |
|---|---|---|---|
| Our algorithm | Chair | 0.0723 | 0.0156 |
| | Bride | 0.1098 | 0.0156 |
| | Flowers | 0.1276 | 0.0156 |
| | Hail | 0.0871 | 0.0156 |
| | **Average** | **0.0992** | **0.0156** |
| Ref. [38] | Pepper | | 0.1650 |
| Ref. [4] | Lena | | 0.0156 |

values by analyzing their combinations in a structured tabular form. The mathematical expression for homogeneity is given in [12].

$$H = \sum \frac{p(r,s)}{1 + |s - r|} \tag{8}$$

Here, $p(r,s)$ represents the pixel value at position $(r,s)$. For enhanced security in the encryption algorithm, lower homogeneity values are desirable. Table 7 presents the homogeneity analysis results for the proposed scheme. The obtained average value of 0.3887 demonstrates improved performance compared to 0.4110 reported in [11].

## 6.7 Statistical analysis

Under this analysis, histogram and correlation coefficient of encrypted images are discussed.

**6.7.1 Cartesian histogram analysis.** A histogram represents the frequency distribution of pixels in a given image. Typically, histograms of the original images exhibit a skewed distribution, while those of encrypted images display a more uniform pattern. This uniformity and smoothness in the histogram bars provide a strong defense against potential histogram-based attacks by adversaries. Figs 26 and 27 illustrate the histograms of both the original and encrypted Chair images.

Observe that the histogram of the original Chair image displays an upward and downward slant. In contrast, the histogram of the encrypted Chair image is notably uniform. This uniformity signifies the robust security attributes of the proposed image encryption algorithm.

The evenness of the histogram is commonly quantified using the following equation, as outlined in [42].

$$\chi^2 = \sum_{i=0}^{255} \frac{(f_i - g_i)^2}{g_i} \tag{9}$$

**Table 7. Homogeneity analysis and comparative evaluation of the proposed scheme.**

| Encryption algorithm | Image | result |
|---|---|---|
| Proposed | Chair | 0.3977 |
| | Bride | 0.3943 |
| | Flowers | 0.3707 |
| | Hail | 0.3922 |
| | **Average** | **0.3887** |
| Ref. [11] | Pepper | 0.4110 |

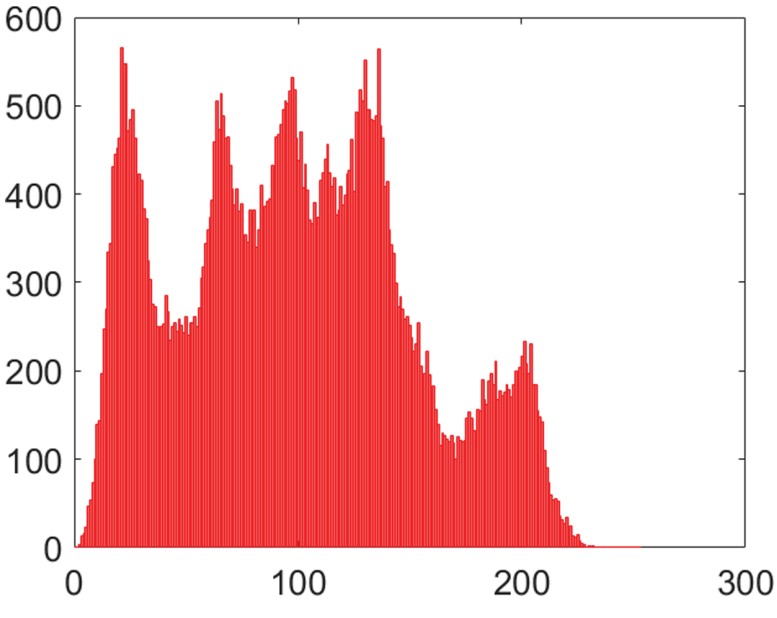

**Fig 26. Histogram of chair plain image.**

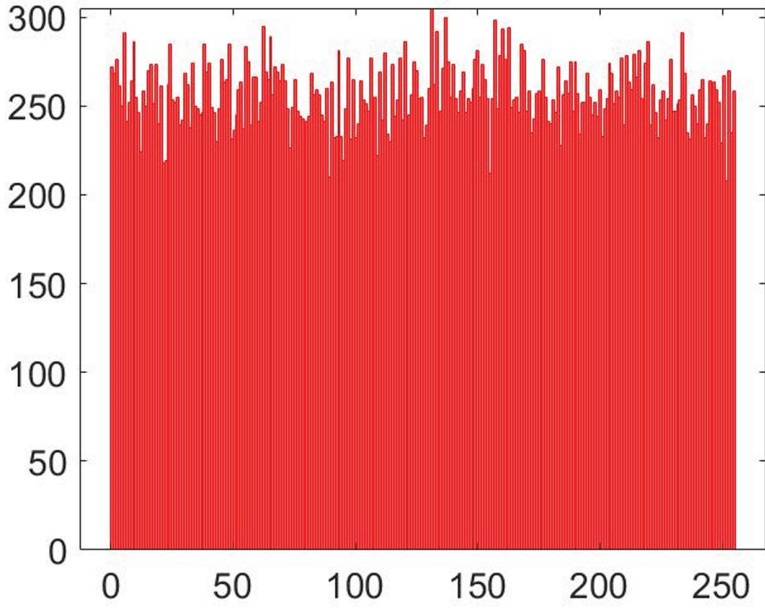

**Fig 27. Histogram of Chair cipher image.**

In the above equation, $f_i$ and $g_i$ denote the observed value and the expected pixel value respectively. If $\chi^2 < \chi^2_{0.05}(255) = 293.24783$, the histogram is considered in the uniform distribution. Table 8 shows the values of $\chi^2$ by the suggested algorithm.

Since the obtained results for the security parameter of $\chi^2$ satisfy the inequality $\chi^2 < \chi^2_{0.05}(255) = 293.24783$, hence we are justified in asserting that the proposed image cipher is immune to the statistical attack.

**Table 8. Results of $\chi^2$ for encrypted and original images.**

| Encryption Algorithms | Images | Original | Encrypted |
|---|---|---|---|
| Proposed | Chair | $1.57681 \times 10^5$ | 253.0087 |
| | Bride | $1.87662 \times 10^5$ | 251.2088 |
| | Flowers | $1.99843 \times 10^5$ | 252.9781 |
| | Hail | $1.90652 \times 10^5$ | 248.6511 |

**6.7.2 Polar histogram analysis.** The polar histogram of the plain image typically exhibits smooth variations with distinct peaks, corresponding to dominant intensity values. This suggests the presence of regions with consistent brightness or similar shading [64].

**Uniform Distribution**: In contrast, the polar histogram of the ciphered image is expected to be more uniform, displaying a rougher texture and a more evenly distributed range of intensity values. This reflects the encryption process, which aims to randomize pixel intensities and eliminate recognizable patterns from the original image.

**Absence of Peaks**: Unlike the plain image, the ciphered image should not contain prominent peaks, as encryption ensures an even redistribution of pixel values across the intensity range. This randomness enhances security by preventing any patterns from being detected.

Fig 28) exhibits a uniform distribution with no distinct peaks, unlike the plain image (Fig 29). This highlights the robustness of the proposed encryption method in effectively obscuring image features.

**6.7.3 Correlation analysis.** A significant connection exists among neighboring pixels within an unencrypted image. When this unencrypted image undergoes encryption, its pixel color codes undergo a comprehensive transformation in both their spatial arrangement and color values. This metric quantifies this transformation. The pixel correlation within the unencrypted image approaches 1, while for the encrypted image, it approaches zero.

To assess this, we randomly selected 3,000 pairs of pixels from unencrypted as well as encrypted images. Following mathematical formula, as outlined in [43], was applied for this evaluation:

$$CC = \frac{P \sum_{d=1}^{P} (z_d \times w_d) - \sum_{d=1}^{P} z_d \times \sum_{d=1}^{P} w_d}{\sqrt{\left(P \sum_{d=1}^{P} z_d^2 - \left(\sum_{d=1}^{P} z_d\right)^2\right)\left(P \sum_{d=1}^{P} w_d^2 - \left(\sum_{d=1}^{P} w_d\right)^2\right)}} \tag{10}$$

In this formula $z$ and $w$ denote the adjacent pixels' intensity values. Moreover, the total number of pixels has been represented by $P$. Apart from that, Figs 30 to 35 show the distribution of pixels in the directions of the horizontal, vertical, and diagonal.

Correlation coefficients between the input plain image and the output encrypted image have been shown in Table 9.

These results are for the image of Chair. According to the entries in Table 9, these results are close to 1 for the plain sample image. Apart from that, these results steeply dropped and reached near zero for the encrypted and cipher images. Jointly, both Figure 8 and Table 3 depict the reality that the values of the correlation coefficient came down dramatically as soon as the encryption algorithm got applied to the plain images. Besides, a comparative analysis has also been made (Table 10). The results of the suggested scheme are comparable with the studies [44,79–81].

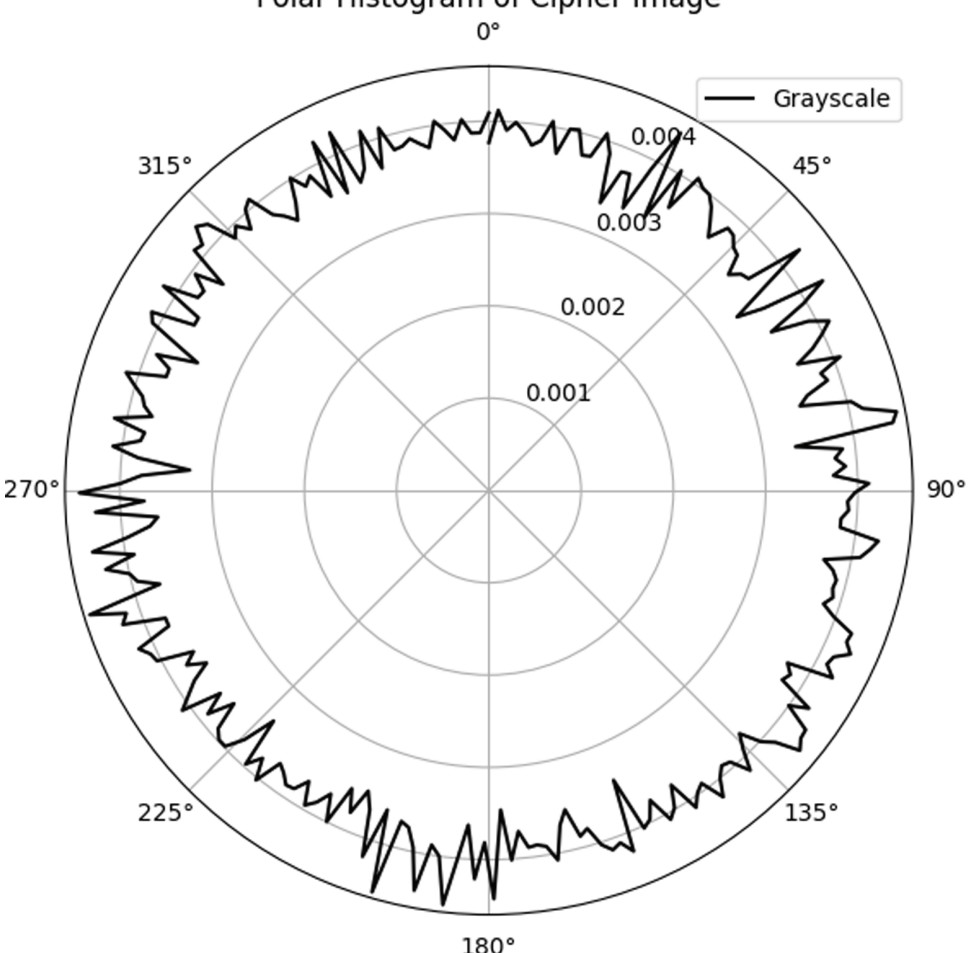

**Fig 28. Polar histogram of Chair cipher image.**

## 6.8 Information entropy

As some image's pixels are disturbed after applying an encryption algorithm over them, its pixels get disarrayed abundantly. To measure it, information entropy is used. Shannon [45], an information theorist, gave its mathematical formula given below:

$$Z(r) = \sum_{c=0}^{2^n-1} p(r_c) log \frac{1}{p(r_c)} \tag{11}$$

Where $Z(r)$ is the information entropy of a signal $r$. In the above equation, $p(r_c)$ corresponds to the probability of $r_c$. For an image whose pixels have been randomized ideally with gray picture values of 256, its value is 8.

Table 11 shows the selected picture entropies and the overall results are nearly equal to 8. In this situation, it is justified to say that envisioned scheme is strong as the entropy is concerned. Besides, Table 11 has also performed a comparison between the published research studies and the current study. Our work beats the work [80].

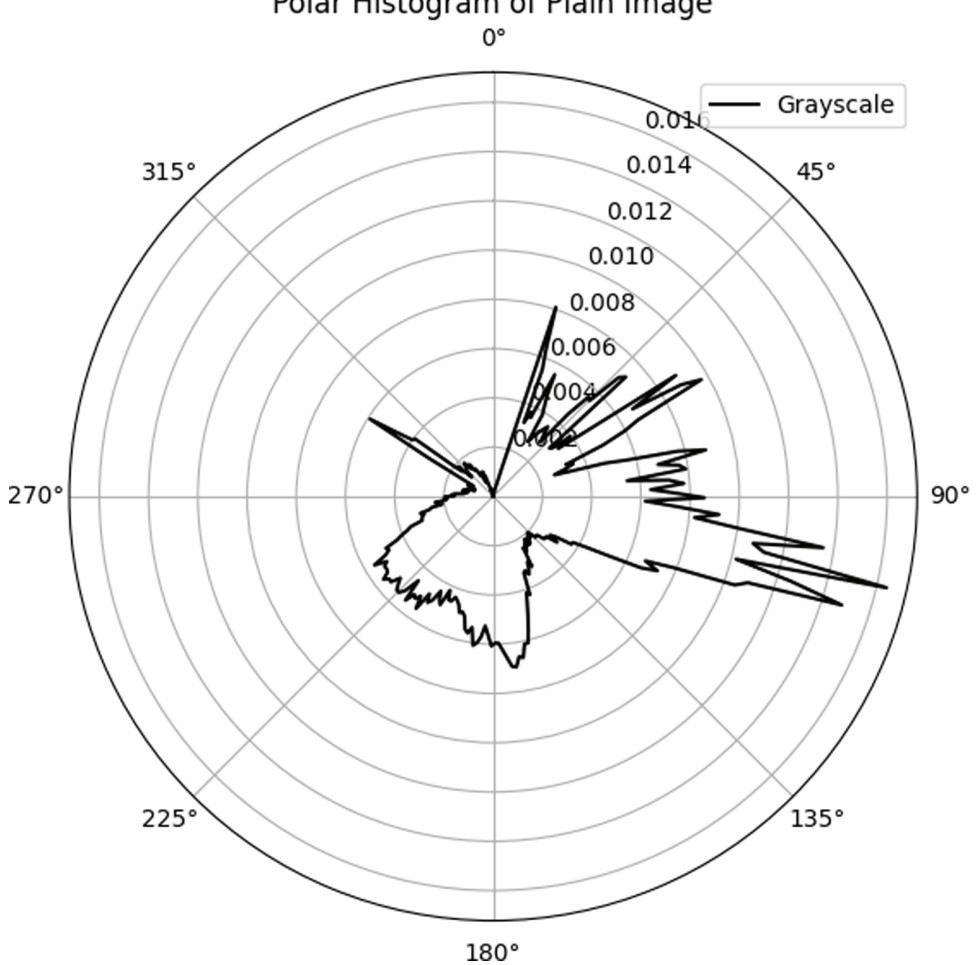

**Fig 29. Polar histogram of Chair plain image.**

## 6.9 Local Shannon entropy (LSE)

Local Shannon entropy is another security parameter for appraising the randomness present in the cipher images. Local-Shannon-entropy can be mathematically represented as [40,41].

$$\overline{S_{l,U_E}(H)} = \sum_{j=1}^{l} \frac{S(H_j)}{l} \tag{12}$$

In this equation $H_1, H_2, H_3, ..., H_l$ correspond to the non-overlapping blocks with $U_E$ pixels selected arbitrarily in the given cipher image. The $S(H_j)(j = 1, 2, 3, ..., l)$ have been gotten through the equation (6). In experimentation set-up, we assigned these values $l = 30, U_E = 1936$. It means that 1936 pixels were chosen for all the 30 blocks against the encrypted image. For the $\alpha$ – level confidence of 0.05, obtained local-Shannon-entropy result comes in between 7.901901305 to 7.903037329 as the study [39] indicates. Table 12 renders the local Shannon entropy results for the given images. One can see that the results of all the images successfully fell in the required threshold. It indicates that the proposed image cipher is furnished with good security effects.

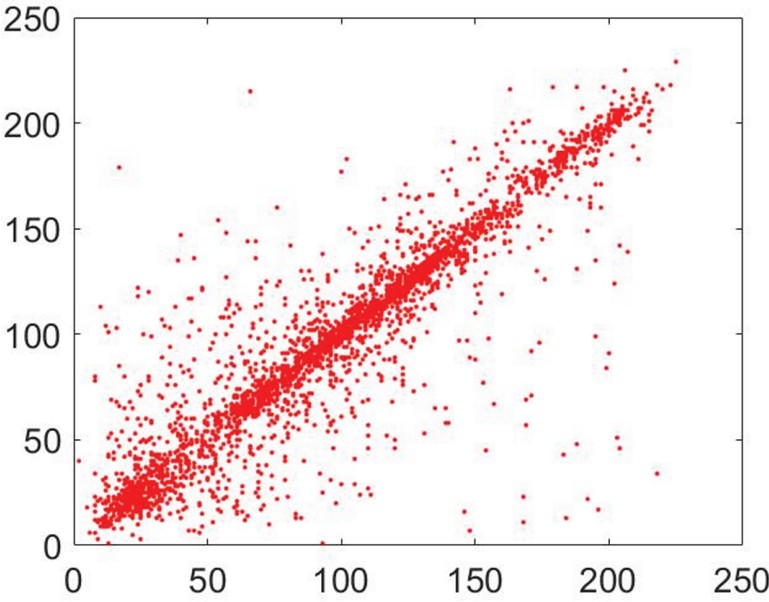

**Fig 30. Correlation distribution for the horizontal orientation of the plain Chair image.**

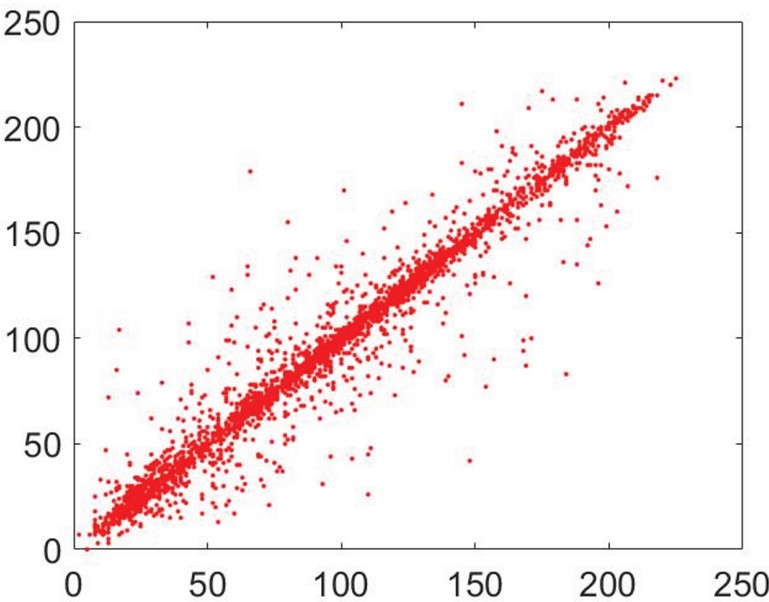

**Fig 31. Correlation distribution for the vertical orientation of the plain Chair image.**

## 6.10 Anti-jamming capability analysis

Robust image encryption systems inherently possess the ability to withstand various potential attacks. Among these, noise and data loss attacks pose significant threats to image ciphers. To evaluate the resilience of our proposed encryption scheme, we selected the cipher images of Chair and Bride.

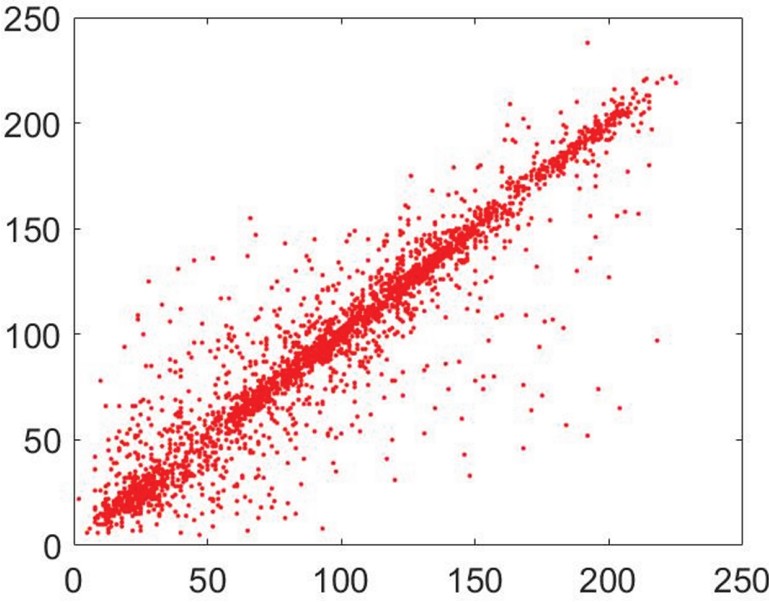

**Fig 32. Correlation distribution for the diagonal orientation of the plain Chair image.**

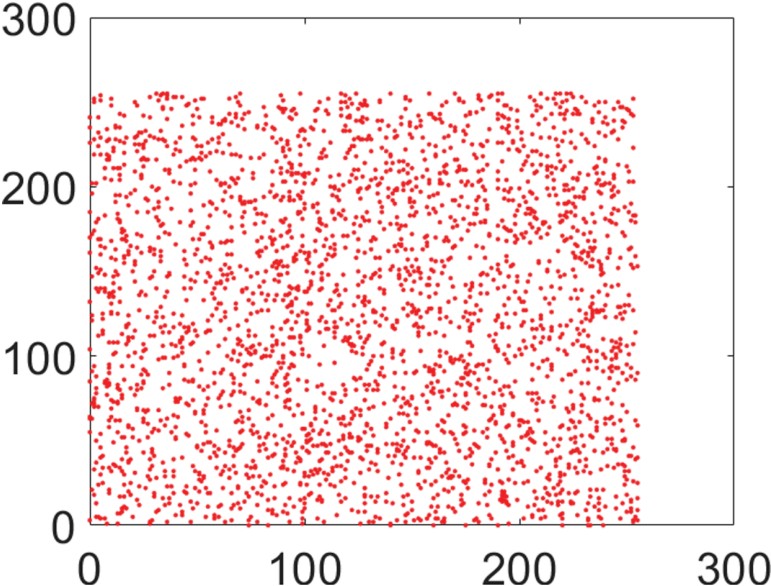

**Fig 33. Correlation distribution for the horizontal orientation of the cipher Chair image.**

To simulate real-world adversarial conditions, we artificially introduced salt-and-pepper noise into the cipher image of Chair (Fig 36). Additionally, we deliberately discarded the upper half of the cipher image of Bride (Fig 38). We then applied our decryption algorithm to these altered images one by one. The restored images, shown in Figs 37 and 39, remain easily recognizable.

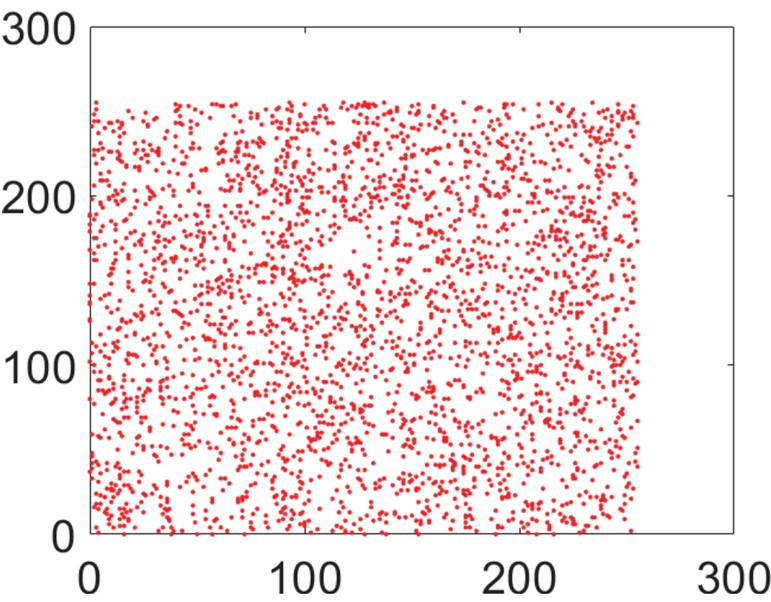

**Fig 34. Correlation distribution for the vertical orientation of the cipher Chair image.**

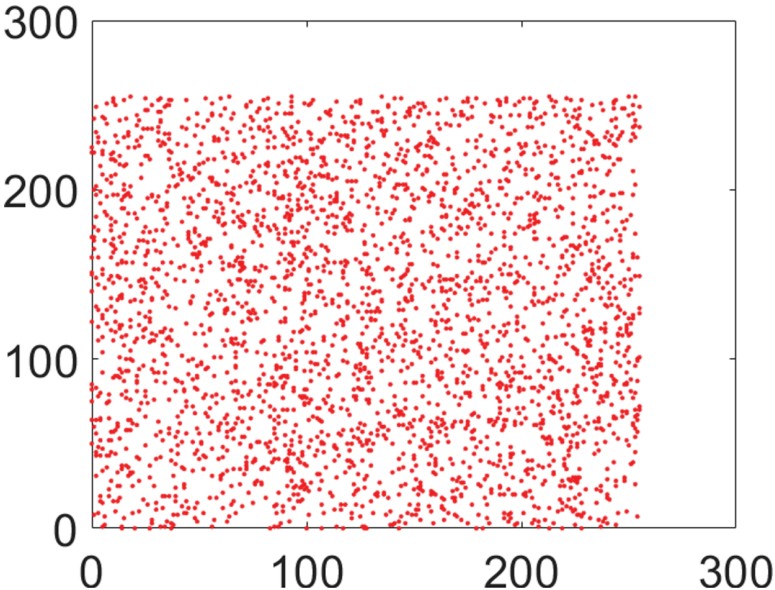

**Fig 35. Correlation distribution for the diagonal orientation of the cipher Chair image.**

This experiment demonstrates that the proposed encryption method effectively withstands noise and data cropping attacks, reinforcing its robustness against potential threats.

### 6.11 Avalanche (differential) attack analysis

Adversaries and malicious actors possess a high level of ingenuity when it comes to identifying vulnerabilities and exploiting them for their malicious intentions. In this specific attack

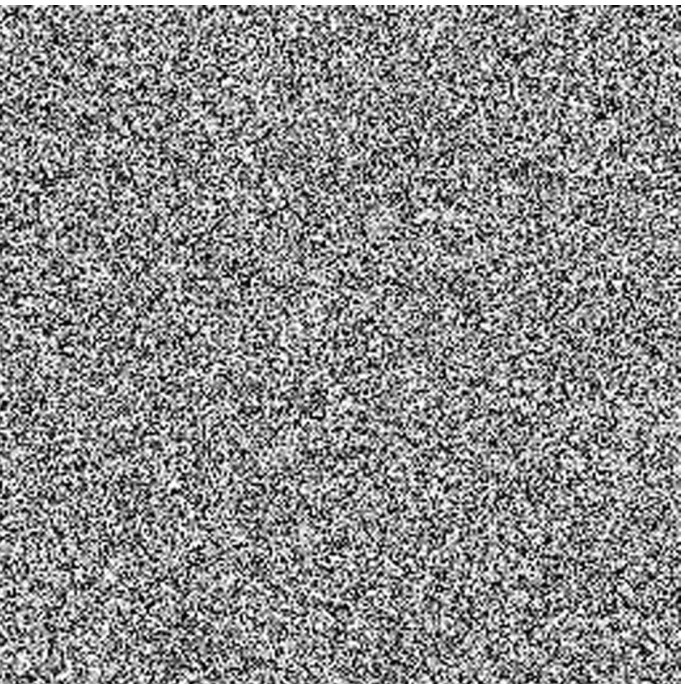

**Fig 36. Pepper & Salt noise with density of 0.1 in the Chair image.**

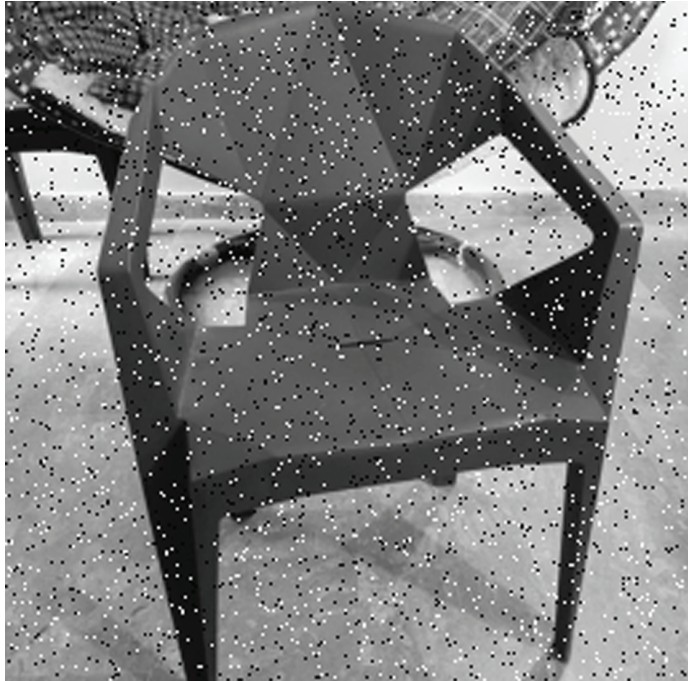

**Fig 37. Decrypted Chair image from Fig 36.**

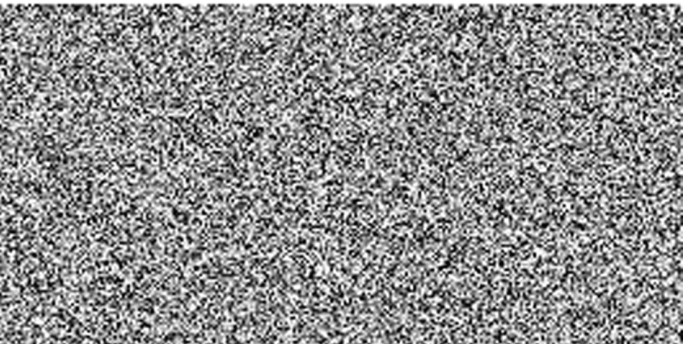

**Fig 38. Bride image with a data crop attack of $1/2 \times m \times n$.**

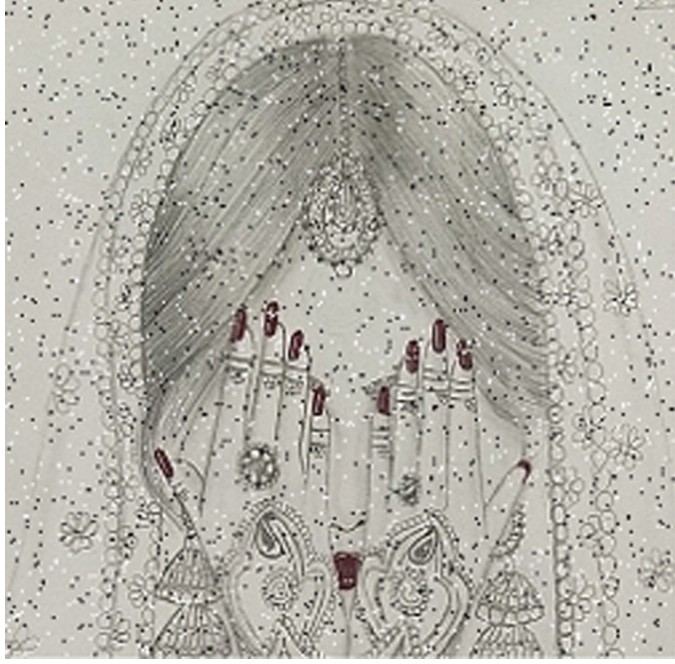

**Fig 39. Decrypted bride image from Fig 38.**

**Table 9. The Correlation coefficient between plain and cipher images.**

| Image | Correlation direction | | |
|---|---|---|---|
| | Horizontal | Vertical | Diagonal |
| Original Chair Image | 0.9223 | 0.9217 | 0.9198 |
| Encrypted Chair image | 0.0076 | -0.0080 | 0.0061 |

**Table 10. Comparison of correlation coefficients of adjacent pixels using different encryption methods.**

| Image | Encryption Algorithm | Correlation direction | | |
|---|---|---|---|---|
| | | Horizontal | Vertical | Diagonal |
| Original Chair image | | 0.9223 | 0.9217 | 0.9198 |
| Encrypted Chair image | Suggested | 0.0076 | -0.0080 | 0.0061 |
| | Ref. [44] | -0.0063 | 0.0065 | -0.0016 |
| | Ref. [80] | 0.0005 | 0.1313 | -0.0047 |
| | Ref. [79] | 0.0013 | -0.0009 | -0.0023 |
| | Ref. [81] | 0.0033 | 0.0070 | 0.0027 |

**Table 11. Results of entropy.**

| Encryption Algorithms | Images | Original | Encrypted |
|---|---|---|---|
| Proposed | Chair | 7.5954 | 7.9974 |
| | Bride | 6.9730 | 7.9978 |
| | Flowers | 7.0097 | 7.9975 |
| | Hail | 7.3424 | 7.9975 |
| | **Average** | 7.2301 | 7.9976 |
| Ref. [44] | Lena | 7.5954 | 7.9978 |
| Ref. [80] | Baboon | | 7.9966 |
| Ref. [79] | - | | 7.9999 |
| Ref. [81] | Lena | | 7.99918 |

**Table 12. Local Shannon entropy results.**

| Images | Local Shannon Entropy | Result |
|---|---|---|
| Chair | 7.902372 | Pass |
| Bride | 7.902276 | Pass |
| Flowers | 7.902844 | Pass |
| Hail | 7.902087 | Pass |

scenario, they acquire two samples of the plaintext image: one with a subtle alteration in a pixel, and the other as the original, unaltered image. Both of these images are then subjected to the encryption process using the encryption algorithm. A meticulous examination of these two resulting cipher images enables them to deduce the secret key. The security parameters of NPCR (Number of Pixels Change Rate) and UACI (Unified Average Changing Intensity) serve as tools to assess the effectiveness of this attack. These metrics quantify the rate of pixel changes and the unified average changing intensity. The following mathematical formulas are employed for this purpose.

$$NPCR = \frac{\sum_{f,p} T(f,p)}{F \times P} \times 100\% \tag{13}$$

where $F$ and $P$ correspond to image's dimensions. $T(f, p)$ can be defined by:

$$T(f,p) = \begin{cases} 1, & \text{if } R(f,p) \neq R'(f,p) \\ 0, & \text{if } R(f,p) = R'(f,p). \end{cases} \tag{14}$$

$$UACI = \frac{1}{F \times P} \left[ \sum_{f,p} \frac{|R(f,p) - R'(f,p)|}{255} \right] \times 100\% \tag{15}$$

In equation 15, the symbols $R$ and $R'$ depict the encoded images before and after the change is introduced in the image. If the values of NPCR and UACI come out to be close to 100% and 33.3% respectively then the cipher is immune to the potential differential attack threats. Table 13 shows the results against these security parameters. 33.4794% and 99.6211% came out to be the average values for UACI and NPCR respectively. According to these results, we assert that the proposed image cipher has the potential to withstand such kinds of threats from the future hackers. Apart from that, a comparison has also been made with other algorithms [44,79–81] in Table 13. The values 99.6204 and 33.4623 (for Chair image) of metrics NPCR and UACI beat all the four studies given in the Table 13.

Apart from that, Tables 14 and 15 show theoretical (critical) values for NPCR and UACI respectively for the images with different sizes. According to the significance levels of $\alpha = 0.05, \alpha = 0.01, \alpha = 0.001$. $N^*_{0.05}, N^*_{0.01}, N^*_{0.001}$ denote critical values in order to reject null hypothesis (Table 14). It implies if value of NPCR for two cipher images is lower than $N^*_\alpha$ then two cipher images do not contain requisite randomness along with an $\alpha$-level significance. According to the Table 16, NPCR results at all confidence levels (for all sizes) meet critical (theoretical) criterion of NPCR randomness test. For UACI, critical value $U^*_\alpha$ consists of two portions, i.e., left value ($U^{*-}_\alpha$) and right value ($U^{*+}_\alpha$). In Table 17, these values have been calculated. Null hypothesis is rejected if some value falls outside the range of ($U^{*-}_\alpha, U^{*+}_\alpha$). Values of UACI for all the sizes of suggested cipher meet critical values of UACI randomness test according to Table 17.

## 6.12 Ciphertext-only, known plaintext, chosen-plaintext and chosen-ciphertext attacks analysis

Cryptanalysis encompasses numerous techniques aimed at compromising ciphers and gaining unauthorized access to the secret key. Known plaintext, Chosen-plaintext, Ciphertext-only, and Chosen-ciphertext attacks are strategies employed by hackers

**Table 13. Average results of metrics NPCR and UACI.**

| Cipher | Image | NPCR(%) | UACI(%) |
|---|---|---|---|
| Proposed | Chair | 99.6204 | 33.4623 |
| | Bride | 99.6059 | 33.3166 |
| | Flowers | 99.6388 | 33.5185 |
| | Hail | 99.6194 | 33.4723 |
| | **Average** | 99.6211 | 33.4424 |
| Ref. [44] | - | 99.5804 | 33.4533 |
| Ref. [80] | Lena | 99.60 | 36.11 |
| Ref. [79] | - | 99.62 | 33.46 |
| Ref. [81] | Lena | 99.61937 | 33.44153 |

**Table 14. Critical values in percentage for NPCR Randomness test.**

| Size | $N^*_{0.05}$ | $N^*_{0.01}$ | $N^*_{0.001}$ |
|---|---|---|---|
| $256 \times 256$ | 99.5693 | 99.5527 | 99.5341 |
| $512 \times 512$ | 99.5893 | 99.5810 | 99.5717 |
| $1024 \times 1024$ | 99.5994 | 99.5952 | 99.5906 |
| $256 \times 256$ | 99.5693 | 99.5527 | 99.5341 |
| $384 \times 384$ | 99.5827 | 99.5716 | 99.5592 |
| $512 \times 512$ | 99.5893 | 99.5810 | 99.5717 |
| $1024 \times 1024$ | 99.5994 | 99.5952 | 99.5906 |
| $1536 \times 1536$ | 99.6027 | 99.5999 | 99.5968 |
| $3072 \times 3072$ | 99.6060 | 99.6047 | 99.6031 |

**Table 15. Critical values in percentage for UACI Randomness test.**

| Size | $\dfrac{u^{*-}_{0.05}}{u^{*+}_{0.05}}$ | $\dfrac{u^{*-}_{0.05}}{u^{*+}_{0.05}}$ | $\dfrac{u^{*-}_{0.05}}{u^{*+}_{0.05}}$ |
|---|---|---|---|
| $256 \times 256$ | 33.2824 | 33.7016 | 33.7677 |
| | 33.6447 | 33.2254 | 33.1593 |
| $384 \times 384$ | 33.5843 | 33.6222 | 33.6663 |
| | 33.3427 | 33.3048 | 33.2607 |
| $512 \times 512$ | 33.5541 | 33.5825 | 33.6156 |
| | 33.3729 | 33.3445 | 33.3114 |
| $1024 \times 1024$ | 33.5088 | 33.5230 | 33.5395 |
| | 33.4182 | 33.4040 | 33.3875 |
| $1536 \times 1536$ | 33.4937 | 33.5032 | 33.5142 |
| | 33.4333 | 33.4238 | 33.4128 |
| $3072 \times 3072$ | 33.4786 | 33.4833 | 33.4888 |
| | 33.4484 | 33.4437 | 33.4382 |

**Table 16. NPCR Randomness test in comparison with theoretical (critical) values.**

| Image size | NPCR value of suggested scheme | 0.05- level | 0.01- level | 0.001- level |
|---|---|---|---|---|
| $256 \times 256$ | 99.6156 | Pass | Pass | Pass |
| $384 \times 384$ | 99.6133 | Pass | Pass | Pass |
| $512 \times 512$ | 99.6192 | Pass | Pass | Pass |
| $1024 \times 1024$ | 99.6129 | Pass | Pass | Pass |
| $1536 \times 1536$ | 99.6145 | Pass | Pass | Pass |
| $3072 \times 3072$ | 99.6181 | Pass | Pass | Pass |

**Table 17. UACI Randomness test in comparison with theoretical (critical) values.**

| Image size | UACI value of suggested scheme | $\dfrac{u^{*-}_{0.05}}{u^{*+}_{0.05}}$ | $\dfrac{u^{*-}_{0.05}}{u^{*+}_{0.05}}$ | $\dfrac{u^{*-}_{0.05}}{u^{*+}_{0.05}}$ |
|---|---|---|---|---|
| $256 \times 256$ | 33.5612 | Pass | Pass | Pass |
| $384 \times 384$ | 33.5192 | Pass | Pass | Pass |
| $512 \times 512$ | 33.5201 | Pass | Pass | Pass |
| $1024 \times 1024$ | 33.4698 | Pass | Pass | Pass |
| $1536 \times 1536$ | 33.4723 | Pass | Pass | Pass |
| $3072 \times 3072$ | 33.4612 | Pass | Pass | Pass |

occasionally [5,6,46]. In a ciphertext-only attack, attackers have limited ciphertexts at their disposal. For the known plaintext attack, they possess pairs of ciphertexts and plaintexts. Chosen plaintext attacks involve full access to the encryption algorithm, allowing attackers to transform given plaintexts into their ciphertext equivalents.

In this context, if experiments confirm the resilience of the proposed cipher to chosen plaintext attacks, then resistance to ciphertext-only, known plaintext, and chosen-ciphertext attacks is inherently assured, as the latter attacks are subsets of the chosen plaintext attack. In a chosen plaintext attack scenario, where attackers have access to the encryption machinery, they manipulate a special image, such as the White image (Fig 40), to derive the cipher's secret key. When this White image is encrypted (Fig 41), it retrieves random data within the encryption algorithm, consequently revealing the secret key.

After these experiments, attackers take another plaintext image, let's say the Bride image, and encrypt it (Fig 42) through the known plaintext attack, utilizing the secret key employed in the encryption of the White image. As Fig 43 illustrates, the Bride cipher image remains secure with this secret key. This experiment is similarly demonstrated using another special image, Black (Figs 44 to 47). These findings unequivocally demonstrate the proposed image cipher's resistance to the aforementioned attacks.

## 6.13 PSNR and SSIM analyses

*PSNR*, an abbreviation for Peak Signal-to-Noise Ratio, serves as an additional security evaluation measure. It quantifies the degree of dissimilarity between plaintext and ciphertext images. Mathematically, it is expressed as follows [60]

$$
\begin{cases}
PSNR = 20log_{10}\left(255/\sqrt{MSE}\right)dB \\
MSE = \frac{1}{M \times N} \sum_{i=1}^{M} \sum_{j=1}^{N} \left(P_0(i,j) - P_1(i,j)\right)^2
\end{cases}
\tag{16}
$$

Image dimensions are denoted as $(M, N)$ in the equation above. Additionally, $P_0(i,j)$ and $P_1(i,j)$ represent pixel intensity values for plaintext and ciphertext images, while *MSE*

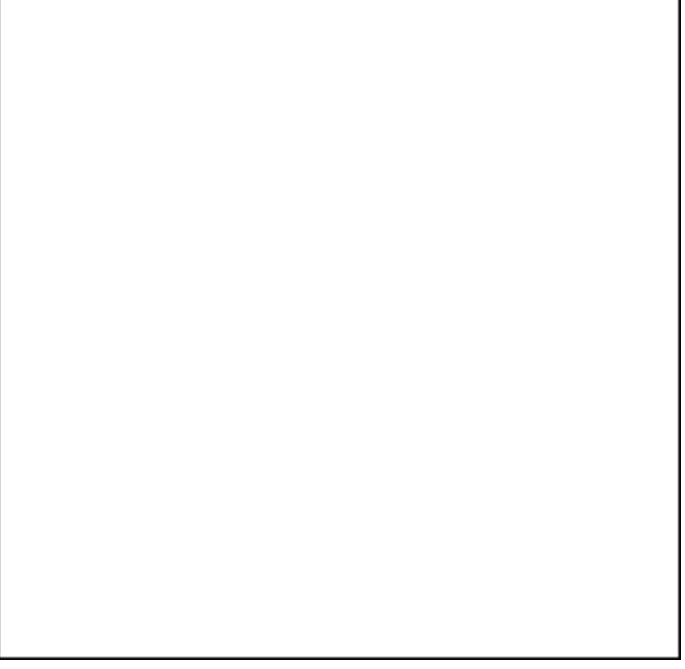

**Fig 40. Original White image.**

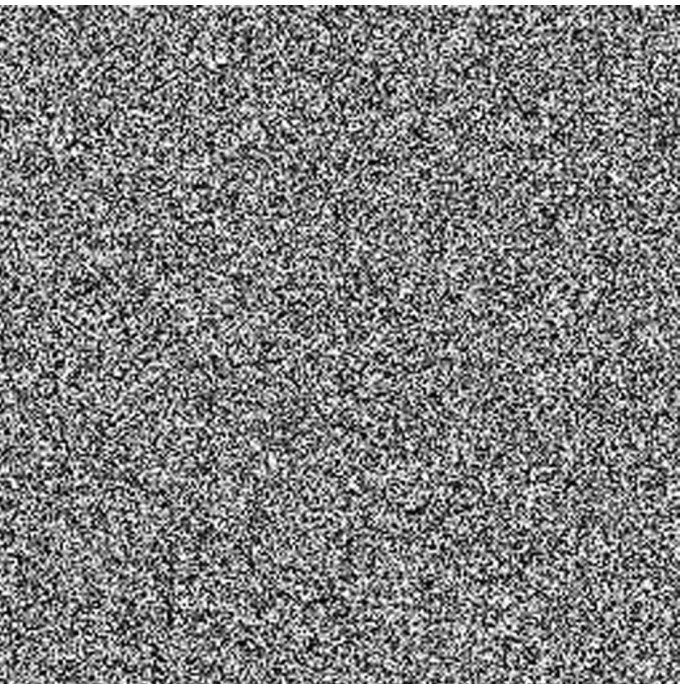

**Fig 41. Cipher White image.**

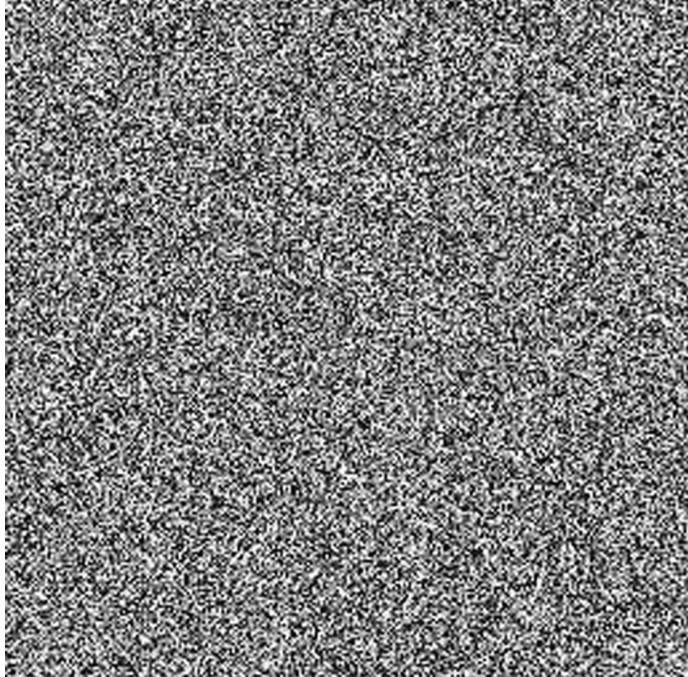

**Fig 42. Cipher chair image.**

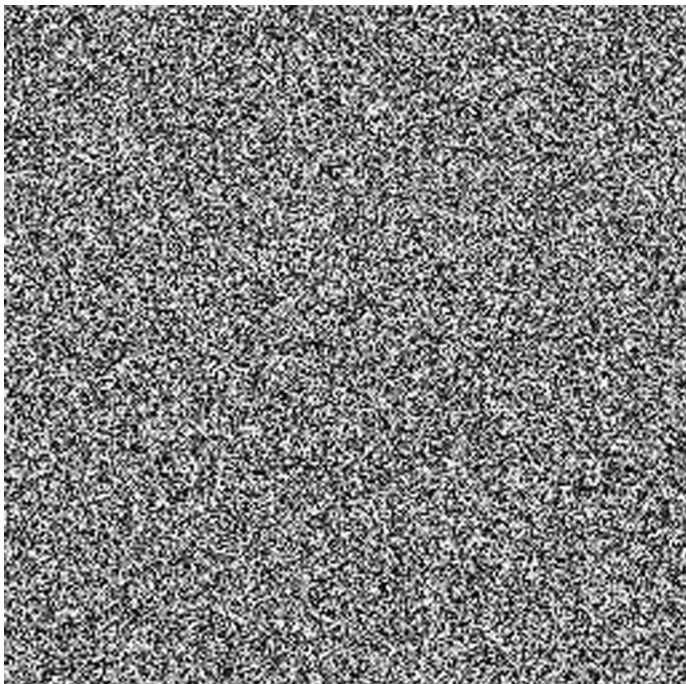

**Fig 43. Decoded image of Chair along with the probable secret key from White image.**

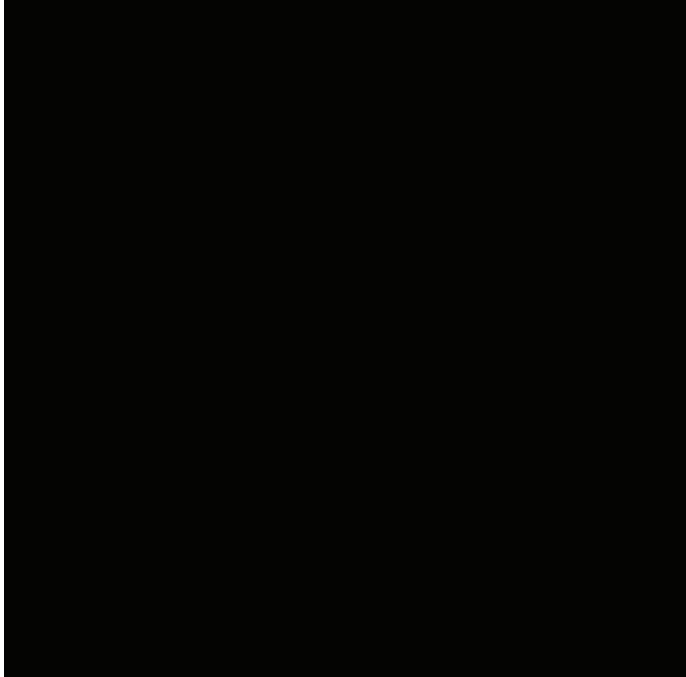

**Fig 44. Original Black image.**

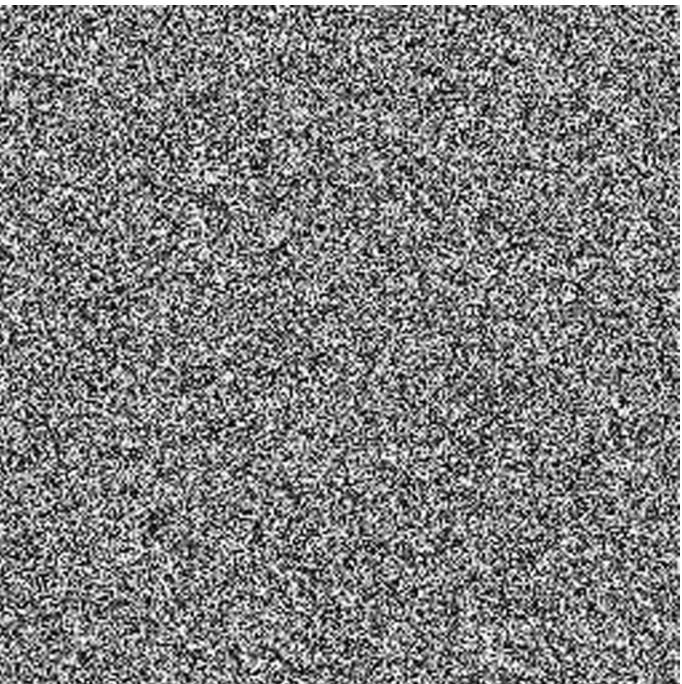

**Fig 45. Cipher Black image.**

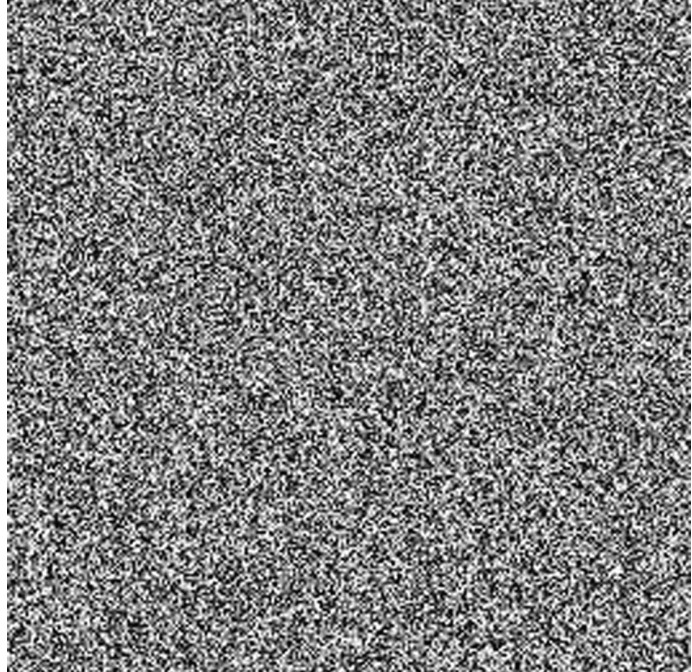

**Fig 46. Cipher Chair image.**

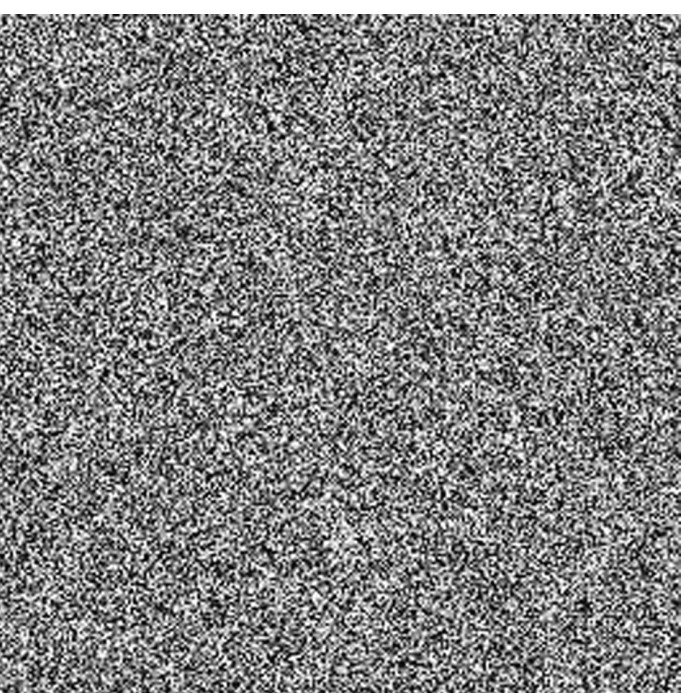

**Fig 47. Decoded image of Chair along with the probable secret key from Black image.**

quantifies mean squared error between these two images. For strong security outcomes, it is desirable to have larger values for *MSE* and smaller values for *PSNR*. Table 18 displays the results of this metric for various techniques.

Notably, the metric attains an infinite ($\infty$) value for original and decrypted images, indicating their absolute similarity due to *MSE* = 0. This underscores that the suggested algorithm is non-lossy. Furthermore, our algorithm achieves a superior *PSNR* value for the image of Chair compared to those in [59,60], signifying a higher level of similarity between plaintext and ciphertext images.

For appreciating the quality of encryption, structural similarity index (SSIM) is also sometimes employed by the cryptographers. SSIM provides a more consistent measure of similarity between encrypted and decrypted images compared to MSE. In other words, it aligns with the concept that a blurred image is of poor quality, making it more compatible with the security parameters of SSIM than with MSE. Calculating SSIM involves considering the standard deviation, mean, and cross-correlation values for plaintext image (P) and output image(E). Formula for SSIM is detailed in [61].

$$SSIM = \frac{(2\bar{P}\bar{E} + d_1)(2\delta_{PE} + d_2)}{(\bar{P}^2 + \bar{E}^2 + d_1)(\delta_P^2 + \delta_E^2 + d_2)} \tag{17}$$

where

$$\bar{P} = \frac{1}{K \times Q} \sum_{k=1}^{K} \sum_{q=1}^{Q} P(k,q), \tag{18}$$

**Table 18. The PSNR & SSIM results: "O-C" represents the original and cipher images, and "O-D" denotes the original and decrypted images.**

|  | Metric | Chair | Bride | Flowers | Hail |
|---|---|---|---|---|---|
| Ours | PSNR (O-D) | ∞ | ∞ | ∞ | ∞ |
|  | PSNR (O-C) | 8.7387 | 8.7412 | 8.6419 | 8.7088 |
| Ref. [58] | PSNR (O-C) | 8.6878 |  |  |  |
| Ref. [59] | PSNR (O-C) | 9.0486 |  |  |  |
| Ref. [60] | PSNR (O-C) | 9.2736 |  |  |  |
| Ours | SSIM | $-1.9287 \times 10^{-7}$ | $2.8722 \times 10^{-6}$ | $3.9911 \times 10^{-9}$ | $4.7234 \times 10^{-10}$ |
| Ref. [61] |  | 0.0063 |  |  |  |

$$\bar{E} = \frac{1}{K \times Q} \sum_{k=1}^{K} \sum_{q=1}^{Q} E(k,q), \tag{19}$$

$$\delta_P = \frac{1}{K \times Q} \sum_{k=1}^{K} \sum_{q=1}^{Q} (P(k,q) - \bar{P})^2, \tag{20}$$

$$\delta_E = \frac{1}{K \times Q} \sum_{k=1}^{K} \sum_{q=1}^{Q} (E(k,q) - \bar{E})^2, \tag{21}$$

and

$$\delta_{PE} = \frac{1}{K \times Q} \sum_{k=1}^{K} \sum_{q=1}^{Q} (P(k,q) - \bar{P})(E(k,q) - \bar{E}) \tag{22}$$

Here, the pair $(\bar{E}, \bar{P})$ corresponds to the means of cipher and plain images respectively. Apart from that, an other pair $(\delta_E, \delta_P)$ refers to the standard deviations of cipher and plain images. Lastly,$\delta_{PE}$ denotes the cross-correlation of the given cipher and plain images. The dimensions of the images are denoted as $(K, Q)$. Additionally, the SSIM value, which falls within the range of 0 to 1, is used for assessment. To ensure stability, $d_1 = (k_1 D)^2$ and $d_2 = (k_2 D)^2$ are employed, with $D$ representing the dynamic range of pixel values (in this case, $D = 255$, $k_1 = 0.01$, and $k_2 = 0.03$). SSIM yields a value of 1 for identical images and nearly zero for structurally dissimilar images. At the bottom of Table 18, the SSIM results are presented. These results closely approach the ideal value of zero, indicating the effectiveness of the proposed image cipher. Moreover, our results surpass those in [61], where the SSIM value was reported as 0.0063.

## 6.14 Speed and time complexity analyses

The suggested cipher for images has been written under the MATLAB environment. Moreover, RAM = 8.00 GB, Intel(R) Core(TM) i7-3740QM CPU @ 2.70GHz was chosen for its hardware. Additionally, the system type is 64-bit Operating System, the x64-based processor. The speed of the cipher with the selected images has been shown in Table 19. The average speed for the chosen images has come out to be 1.2232 seconds. Besides, the proposed work beat the work [80]. Moreover, this work further executed the proposed algorithm for 100 test images with varying sizes (from 256 × 256 to 1024 × 1024) whose average time came out to be 3.2780 seconds. The performance of the algorithms is also checked through the time complexity analysis. Algorithm 1 takes the time $O(6mn)$ to spawn the streams of random data. The

**Table 19. Results of encryption speed along with a previous studies.**

| Approach | Image | Size | Speed (sec) |
|---|---|---|---|
| Proposed | Chair | $256 \times 256$ | 1.2566 |
| | Bride | $256 \times 256$ | 1.2212 |
| | Flowers | $256 \times 256$ | 1.1844 |
| | Hail | $256 \times 256$ | 1.2308 |
| | **Average** | - | **1.2232** |
| | Chair | $512 \times 512$ | 2.8760 |
| | Chair | $1024 \times 1024$ | 3.7612 |
| Ref. [44] | Lena | $256 \times 256$ | 3.1143 |
| Ref. [80] | Baboon | $256 \times 256$ | 2.0 |
| Ref. [79] | - | $256 \times 256$ | 0.54 |
| Ref. [81] | Lena | $256 \times 256$ | 0.3493 |

time consuming steps of Algorithm 2 take the time $O(mn + 2m + 2m + 3m) = O(mn)$ after ignoring the lower order terms. In the same fashion, Algorithm 3 takes the time $O(mn)$. After adding all the terms, we get the time complexity of the suggested algorithm as $O(8mn)$ which is better than $O(24mn)$ [54] and $O(24mn)$ [55]. It takes the form of $O(8n^2)$ in case the image is square.

## 7 Conclusion

Varied image encryption schemes abound, as an objective study of the concerned literature indicates. Among these, a small subset of schemes based on scrambled images has been developed. In addition, various algorithms at the pixel level using both 2D and 3D spaces have already been explored. This study proposes a novel image encryption scheme that builds on the notion of a scrambled image, where randomly selected rows and columns of pixels are inserted into arbitrary row and column positions within the scrambled image. This operation is repeated multiple times to introduce the desired scrambling effects. Diffusion is achieved via an XOR operation between the scrambled image and streams of random numbers, generated using the Rössler chaotic system. Comprehensive validation using various security parameters demonstrates that the proposed image cipher offers strong potential for practical applications, particularly in resource-constrained environments. Furthermore, its lightweight structure and efficient execution indicate real-time feasibility in IoT devices, making it a strong candidate for integration into next-generation secure communication systems. As future work, the concept of the scrambled image can be extended into the 3D space to potentially enhance security further.

## Acknowledgments

The authors are grateful to Umm Al-Qura University, Saudi Arabia for supporting this research work.

## Author contributions

**Conceptualization:** Taher M. Ghazal.

**Formal analysis:** Atif Ikram, Abdelrahman H. Hussein.

**Investigation:** Taher M. Ghazal, Atif Ikram, Marwan Ali Albahar.

**Methodology:** Nasreen Zulfiqar, Foziah Gazzawe, Marwan Ali Albahar.

**Project administration:** Atif Ikram.

**Software:** Tauqir Ahmad, Marwan Ali Albahar.

**Supervision:** Ahmad Salman Khan.

**Validation:** Nasreen Zulfiqar, Taher M. Ghazal, Foziah Gazzawe, Ahmad Salman Khan.

**Visualization:** Tauqir Ahmad, Abdelrahman H. Hussein.

**Writing – original draft:** Nasreen Zulfiqar, Abdelrahman H. Hussein, Ahmad Salman Khan.

**Writing – review & editing:** Tauqir Ahmad, Foziah Gazzawe.

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
