## [Decision Letter · Decision Letter 0]

2 Jun 2025

PONE-D-25-11205Reinforcing Image Security: A High-Entropy Scrambling Algorithm Using the Rossler Chaotic SystemPLOS ONE

Dear Dr. Ikram,

Thank you for submitting your manuscript to PLOS ONE. After careful consideration, we feel that it has merit but does not fully meet PLOS ONE’s publication criteria as it currently stands. Therefore, we invite you to submit a revised version of the manuscript that addresses the points raised during the review process.

We look forward to receiving your revised manuscript.

Kind regards,

Serdar Solak

Academic Editor

PLOS ONE

4. We note that Figures 3,5,6 and 10 in your submission contain copyrighted images. All PLOS content is published under the Creative Commons Attribution License (CC BY 4.0), which means that the manuscript, images, and Supporting Information files will be freely available online, and any third party is permitted to access, download, copy, distribute, and use these materials in any way, even commercially, with proper attribution. For more information, see our copyright guidelines: http://journals.plos.org/plosone/s/licenses-and-copyright.

a. You may seek permission from the original copyright holder of Figures 3,5,6 and 10 to publish the content specifically under the CC BY 4.0 license.

Additional Editor Comments:

Please address the reviewers’ suggestions carefully. If any of the recommended articles are not relevant to your study, you may disregard them. It is important to avoid including unnecessary or unrelated citations.

Reviewers' comments:

Reviewer's Responses to Questions

**Comments to the Author**

1. Is the manuscript technically sound, and do the data support the conclusions?

Reviewer #1: Partly

Reviewer #2: Yes

Reviewer #3: Yes

Reviewer #4: Partly

Reviewer #5: Partly

2. Has the statistical analysis been performed appropriately and rigorously? 

Reviewer #1: Yes

Reviewer #2: Yes

Reviewer #3: No

Reviewer #4: Yes

Reviewer #5: No

3. Have the authors made all data underlying the findings in their manuscript fully available?

Reviewer #1: Yes

Reviewer #2: Yes

Reviewer #3: Yes

Reviewer #4: Yes

Reviewer #5: No

4. Is the manuscript presented in an intelligible fashion and written in standard English?

Reviewer #1: Yes

Reviewer #2: Yes

Reviewer #3: Yes

Reviewer #4: No

Reviewer #5: Yes

5. Review Comments to the Author

Reviewer #1: 1) Manuscript employs the 3D Rossler chaotic system for random number generation, but it does not mathematically justify why this system is preferable over other chaotic maps such as Logistic, Henon, or Chen systems. A Lyapunov exponent analysis, bifurcation diagram, and entropy measure comparison should be added to establish its superiority in key sensitivity and randomness properties.

2) Proposed scrambling approach relies on row- and column-based pixel transpositions but lacks a formal cryptographic security proof. The authors should mathematically demonstrate the diffusion and confusion properties using metrics like Shannon entropy, differential uniformity, and avalanche effect.

3) Proposed claim that the encryption scheme offers a large key space is not substantiated with a thorough key sensitivity analysis. A detailed parameter space exploration (i.e., how changes in initial conditions of the chaotic system affect encryption security) and a comprehensive key space growth analysis are required.

4) The paper focuses on encryption speed as a usability metric but fails to analyze computational efficiency, energy consumption, and real-time feasibility on IoT devices.

5) The paper focuses on encryption speed as a usability metric but fails to analyze computational efficiency, energy consumption, and real-time feasibility on IoT devices. Include it in the revised manuscript.

6) The manuscript does not sufficiently compare the proposed encryption technique against state-of-the-art where high entropy based ciphers images like

DNA based work - Enhancing Image Security via Block Cyclic Construction and DNA Based LFSR

Moore automation- An Applied Image Cryptosystem on Moore’s Automaton Operating on δ (qk)/ 2

LFSR based work- Chaos-based medical image encryption scheme using special nonlinear filtering function based LFSR

7) The proposed scheme is tested on standard test images like Lena and Baboon, but real-world image datasets (e.g., medical, satellite, and biometric images) should be included.

Reviewer #2: About the manuscript “Reinforcing Image Security: A High-Entropy Scrambling Algorithm Using the Rossler Chaotic System”, I think this theme is a hot academic topic. Overall, it sounds technically feasible, and the structure of the manuscript is complete. Yet, here are some suggestions which should be taken into account to improve the current version.

1.Abstract and conclusion should be rewritten as they do not effectively summarize the entire paper. For instance, in the abstract, most of the descriptive language is about the background, which confuses the reader. It should primarily introduce the author's technical solution, key points of the method, and the superior characteristics of the proposed method.

2.The title is about the High-Entropy Scrambling Algorithm, but how is High-Entropy reflected? What testing methods are used to verify, or how is it compared with existing algorithms? As far as I know, Scrambling is just a position-based shuffling.

3.The font size of the coordinate axes in Figure 1 is too small, making it difficult to read. In fact, all figures in the paper have similar issues. It is recommended to uniformly modify them, improve the resolution and quality of the figures, making them highly readable.

4.Figure 2 seems to be the algorithm flowchart of the entire paper? But just from this figure, it is confusing and unclear. From the context of the paper, it should be an image encryption algorithm, and shuffling is the main step. It is suggested that all the main encryption steps should be expressed and uniformly drawn in Figure 2. In other words, Figure 2 is the framework diagram of the entire algorithm.

5.Regarding Figure 8, it looks quite novel and interesting. But I am not clear about how this figure is generated, are there any relevant references or websites? Or executable code? It is hoped that the details can be provided as much as possible so that peers have the opportunity to reproduce the related results.

6.Regarding the discussion on cryptanalysis in Section 5.7, it is recommended to refer to some systematic cryptanalysis work, such as:

https://doi.org/10.1016/j.eswa.2024.123748. https://doi.org/10.1016/j.eswa.2023.121514.

From the above cryptanalysis work, it is seen that the current algorithm is difficult to resist chosen plaintext attacks.

7.Moreover, experimental analysis and results discussion are relatively little, and the enrichment is recommended. At the same time, it is suggested that some of the latest and similar work should be discussed in the introduction, such as in the security enhancement design, which has similarities and enlightening value.

DOI :10.3390/math12243917; DOI:10.1016/j.eswa.2024.123190;

DOI: 10.1007/s11071-021-06206-8; 10.1007/s00371-023-02812-2。

8.The current article draft still contains a significant number of typographical and grammatical errors. Additionally, it is recommended to study the writing and expression of higher-level literature, and to carefully check and correct these issues.

In summary, the current draft is unacceptable and cannot be published, thus my review opinion is major revision. Consider the next step after the revisions are made.

Reviewer #3: Work is nice. A new technique based on the rows and columns in the scrambled image, has been developed. But I have few concerns.

1. Manuscript is furnished with language and grammar issues which must be addressed.

2. The advantages that this technique brings, must be described in a very clear way.

3. Homogeneity analysis needs to be added in the security analysis section.

4. NIST random number analysis should be included.

5. Irregular deviation analysis should be added in the analysis section.

6. Introduction section is weak. It is not developing context sufficiently. The related work should be added.

Reviewer #4: This manuscript introduces a new image scrambling algorithm that employs row- and column-based approaches. It involves inserting input image columns into a 2D scrambled image, repeating the process for rows, and enhancing diffusion through an XoR operation with random number streams generated by the 3D Rössler chaotic system. Both simulation and security analysis demonstrate the cipher's resilience to various cryptanalytic attacks, recommending it for real-world applications.

Overall, the presented work exhibits a certain degree of novelty and contribution. However, the manuscript has noticeable deficiencies in several aspects. I strongly recommend that the authors carefully revise and improve the manuscript based on the review comments.

1). The abstract does not meet the general requirements. Please refer to A novel multi-channel image encryption algorithm leveraging pixel reorganization and hyperchaotic maps and Exploiting robust quadratic polynomial hyperchaotic map and pixel fusion strategy for efficient image encryption for guidance on enhancements. The authors should concisely elaborate on the research background in the abstract, identify the research gap, outline the main content of their work (including key steps), present the key findings, and emphasize the superiority and significance of their contributions.

2). It is recommended that the authors make appropriate adjustments to the list of keywords. Typically, we should meticulously choose five to seven keywords that accurately capture the core content and distinctive attributes of the manuscript, while maintaining simplicity to optimize retrievability. Please refer to Exploiting newly designed fractional-order 3D Lorenz chaotic system and 2D discrete polynomial hyper-chaotic map for high-performance multi-image encryption and A robust memristor-enhanced polynomial hyper-chaotic map and its multi-channel image encryption application for further guidance. For instance, in my view, "Image encryption," "Pixel scrambling," and "Secure analysis" should at least be selected as keywords.

3). The current introduction is overly concise and lacks necessary content. The authors should enumerate various emerging image encryption algorithms or schemes, such as those based on DNA computing, quantum computing, optical transformations, chaotic systems, neural networks, compressive sensing, and so forth, and introduce some of the latest advancements in the field of image encryption. They should identify research gaps and discuss the motivations and advantages of choosing specific techniques and methods for image encryption. Some noteworthy related works include Dynamic analysis and implementation of FPGA for a new 4D fractional-order memristive hopfield neural network, An image encryption algorithm based on Tabu search and hyperchaos, A novel multi-channel image encryption algorithm leveraging pixel reorganization and hyperchaotic maps, Exploiting robust quadratic polynomial hyperchaotic map and pixel fusion strategy for efficient image encryption, and Exploiting newly designed fractional-order 3D Lorenz chaotic system and 2D discrete polynomial hyper-chaotic map for high-performance multi-image encryption.

4). Cryptanalysis plays a pivotal role in ensuring the rationality, practicality, and security of image encryption algorithms or schemes. Therefore, it is recommended that the authors dedicate a separate paragraph in the introduction to introduce and discuss some recent advancements in cryptanalysis research related to image encryption, such as Cryptanalysis of an image encryption algorithm using quantum chaotic map and DNA coding, Cryptanalyzing an image cipher using multiple chaos and DNA operations, Cryptanalysis and improvement of the image encryption scheme based on Feistel network and dynamic DNA encoding, and Cryptanalysis and improvement of the image encryption scheme based on 2D Logistic-Adjusted-Sine map. Furthermore, a rigorous validation of the proposed image encryption scheme's rationality, practicality, and security should be conducted in light of these cryptanalysis studies.

5). To strengthen the clarity and impact of the manuscript, I suggest that the author concisely highlight the significance and contributions of their work at the end of the introduction, utilizing a bullet-point list format. Please refer to A novel multi-channel image encryption algorithm leveraging pixel reorganization and hyperchaotic maps and Exploiting robust quadratic polynomial hyperchaotic map and pixel fusion strategy for efficient image encryption for guidance on revisions.

6). The manuscript necessitates refinement in the presentation and elucidation of the equations and mathematical symbols. For improvements, please kindly refer to Exploiting newly designed fractional-order 3D Lorenz chaotic system and 2D discrete polynomial hyper-chaotic map for high-performance multi-image encryption and A secure and efficient image transmission scheme based on two chaotic maps.

7). In Section 3, the organization of the content appears somewhat disorganized, lacking necessary details and adequate descriptions. Please refer to A novel multi-channel image encryption algorithm leveraging pixel reorganization and hyperchaotic maps and Exploiting robust quadratic polynomial hyperchaotic map and pixel fusion strategy for efficient image encryption for guidance on improvements. Furthermore, for each encryption step, the authors are requested to clearly articulate the design background, rationale, motivation, process, or decisions underlying it. Otherwise, it may give the impression that the authors are merely arbitrarily stacking encryption steps without due consideration for the rationality, practicality, security, and efficiency of the design.

8). Many figures in the manuscript, such as Figure 2, are of extremely poor quality, containing evident errors and deficiencies. Clearly, they do not meet the quality standards expected of a rigorous scientific research paper. It is recommended that the authors refer to relevant high-quality papers for improvements.

9). In the experimental section, the authors should clearly specify the hardware and software configurations, as well as the benchmark datasets used for their experiments. Please refer to A secure and efficient image transmission scheme based on two chaotic maps and Exploiting robust quadratic polynomial hyperchaotic map and pixel fusion strategy for efficient image encryption for improvements. This will help ensure the evaluability and reproducibility of their experimental results.

10). In Section 4, the authors have not conducted a systematic and comprehensive security evaluation and efficiency analysis of the proposed encryption scheme. Please refer to relevant high-quality papers to improve the details, data, analysis, and discussion for each evaluation item. Furthermore, the authors should compare their proposed encryption scheme with some of the latest encryption schemes to highlight its advantages.

11). While considering security, encryption efficiency is also an important factor that researchers must consider. Therefore, it is recommended that the authors analyze and discuss the encryption efficiency of the proposed image encryption algorithm. Please refer to A novel multi-channel image encryption algorithm leveraging pixel reorganization and hyperchaotic maps and A secure and efficient image transmission scheme based on two chaotic maps. Besides, it is recommended to conduct extensive experiments using test images of different sizes (such as 256×256, 512×512, and 1024×1024) to enrich the experimental data. In addition, the obtained experimental data should be compared with some state-of-the-art encryption algorithms or schemes.

12). The conclusion section does not meet the requirements. Please kindly refer to A novel multi-channel image encryption algorithm leveraging pixel reorganization and hyperchaotic maps and Exploiting robust quadratic polynomial hyperchaotic map and pixel fusion strategy for efficient image encryption for improvements. The authors should present current problems, solutions, contributions, findings, limitations, and prospects for future work in a more engaging manner.

13). The authors are strongly advised to adjust their reference list. They should remove outdated references as much as possible and cite, introduce, analyze, discuss, and compare the latest related works, especially works related to chaotic cryptography. Outdated or low-quality references should be removed. This will help to highlight the importance, significance, superiority, and timeliness of their work.

14). The linguistic quality of the manuscript is suboptimal. The authors are recommended to meticulously proofread the manuscript to enhance the logicality, rigor, and consistency of the content.

Reviewer #5: I have attached a file titled 'PONE-D-25-11205.pdf' for the authors' reference, which contains detailed comments and suggestions aimed at improving the manuscript. The document includes specific feedback on various aspects of the work, such as the methodology, experimental validation, and overall presentation.

6. PLOS authors have the option to publish the peer review history of their article (what does this mean?). If published, this will include your full peer review and any attached files.

Reviewer #1: No

Reviewer #2: No

Reviewer #3: **Yes: **Dr. Nadeem Iqbal

Reviewer #4: No

Reviewer #5: No

---

## [Author Response · Author response to Decision Letter 1]

3 May 2025

Response to the reviewers has already been uploaded as a separate file.

---

## [Decision Letter · Decision Letter 1]

27 May 2025

Securing Digital Images: A Chaos-Driven Scrambling Algorithm Using the Rossler System¨

PONE-D-25-11205R1

Dear Dr. Ikram,

We’re pleased to inform you that your manuscript has been judged scientifically suitable for publication and will be formally accepted for publication once it meets all outstanding technical requirements.

Kind regards,

Serdar Solak

Academic Editor

PLOS ONE

Additional Editor Comments (optional):

Reviewers' comments:

Reviewer's Responses to Questions

**Comments to the Author**

1. If the authors have adequately addressed your comments raised in a previous round of review and you feel that this manuscript is now acceptable for publication, you may indicate that here to bypass the “Comments to the Author” section, enter your conflict of interest statement in the “Confidential to Editor” section, and submit your "Accept" recommendation.

Reviewer #1: All comments have been addressed

Reviewer #3: All comments have been addressed

Reviewer #4: All comments have been addressed

2. Is the manuscript technically sound, and do the data support the conclusions?

Reviewer #1: Yes

Reviewer #3: Yes

Reviewer #4: Yes

3. Has the statistical analysis been performed appropriately and rigorously? 

Reviewer #1: Yes

Reviewer #3: Yes

Reviewer #4: Yes

4. Have the authors made all data underlying the findings in their manuscript fully available?

Reviewer #1: Yes

Reviewer #3: Yes

Reviewer #4: Yes

5. Is the manuscript presented in an intelligible fashion and written in standard English?

Reviewer #1: Yes

Reviewer #3: Yes

Reviewer #4: Yes

6. Review Comments to the Author

Reviewer #1: The authors have thoroughly and effectively addressed all the reviewers' comments and suggestions with clear justifications and appropriate revisions throughout the manuscript. Based on the improvements made and the overall quality of the work, I recommend the paper for acceptance.

Reviewer #3: All of my concerns have been addressed in the revised manuscript. So, I think paper can be accepted now. It is better than the previous version.

Reviewer #4: Dear authors, thank you for your revisions. Since my concerns are addressed, I am pleased to recommend acceptance of your manuscript.

7. PLOS authors have the option to publish the peer review history of their article (what does this mean?). If published, this will include your full peer review and any attached files.

Reviewer #1: No

Reviewer #3: No

Reviewer #4: No

---

## [Editor Report · Acceptance letter]

PONE-D-25-11205R1

PLOS ONE

Dear Dr. Ikram,

I'm pleased to inform you that your manuscript has been deemed suitable for publication in PLOS ONE. Congratulations! Your manuscript is now being handed over to our production team.

Kind regards,

on behalf of

Assoc. Prof. Serdar Solak

Academic Editor

PLOS ONE